# An Overview of Stimuli-Responsive Intelligent Antibacterial Nanomaterials

**DOI:** 10.3390/pharmaceutics15082113

**Published:** 2023-08-09

**Authors:** Jinqiao Zhang, Wantao Tang, Xinyi Zhang, Zhiyong Song, Ting Tong

**Affiliations:** 1Hunan Key Laboratory of Economic Crops Genetic Improvement and Integrated Utilization, School of Life and Health Sciences, Hunan University of Science and Technology, Xiangtan 411201, China; q1811346062@163.com (J.Z.); zhangxy@mail.hnust.edu.cn (X.Z.); 2School of Materials Science and Engineering, Zhengzhou University, Zhengzhou 450001, China; tangwt2020@nanoctr.cn; 3College of Science, Huazhong Agricultural University, Wuhan 430070, China

**Keywords:** stimuli-responsive, intelligent antibacterial, synergistic antibacterial, bacterial resistance, bacterial biofilm, antibacterial nanomaterials

## Abstract

Drug-resistant bacteria and infectious diseases associated with biofilms pose a significant global health threat. The integration and advancement of nanotechnology in antibacterial research offer a promising avenue to combat bacterial resistance. Nanomaterials possess numerous advantages, such as customizable designs, adjustable shapes and sizes, and the ability to synergistically utilize multiple active components, allowing for precise targeting based on specific microenvironmental variations. They serve as a promising alternative to antibiotics with diverse medical applications. Here, we discuss the formation of bacterial resistance and antibacterial strategies, and focuses on utilizing the distinctive physicochemical properties of nanomaterials to achieve inherent antibacterial effects by investigating the mechanisms of bacterial resistance. Additionally, we discuss the advancements in developing intelligent nanoscale antibacterial agents that exhibit responsiveness to both endogenous and exogenous responsive stimuli. These nanomaterials hold potential for enhanced antibacterial efficacy by utilizing stimuli such as pH, temperature, light, or ultrasound. Finally, we provide a comprehensive outlook on the existing challenges and future clinical prospects, offering valuable insights for the development of safer and more effective antibacterial nanomaterials.

## 1. Introduction

In recent years, a series of infectious diseases caused by bacteria and bacterial biofilms, including pneumonia, meningitis, sepsis, dental caries, and chronic infections, has been widely recognized as a significant threat to global public health and safety [1,2]. Since the introduction of penicillin as the first antibiotic for clinical treatment in 1923, antibiotics are widely utilized in clinical practice for the treatment of bacterial and biofilm infections, saving numerous lives [3]. While antibiotics have brought tremendous benefits to humanity, their continued misuse and abuse have accelerated the dissemination of antibiotic-resistant strains, leading to adverse drug reactions, as well as organ and central nervous system damage during the metabolic process. The prevalence of infections attributed to drug-resistant bacteria is escalating, with a simultaneous increase in the severity of the drug resistance crisis [4]. Prolonged or excessive use of broad-spectrum antibiotics can suppress or eradicate beneficial normal bacterial populations, causing dysbiosis and weakening the immune system. Many pathogenic bacteria exhibit intrinsic insensitivity to antibiotic treatment or progressively acquire resistance to various antibiotics, resulting in a significant decrease in treatment effectiveness or even ineffectiveness [5]. Moreover, the inappropriate use of antibiotics has led to the emergence of an alarming number of “superbugs” that exhibit multidrug resistance, presenting a significant risk to human health [6,7]. Superbugs that are resistant to antibiotics will seriously reduce the effectiveness of people’s antibacterial methods based on antibiotics and lead to a public health crisis. Many countries are consciously raising public awareness of antibiotics, strictly regulating the scope and process of antibiotic use, and preventing the misuse or improper use of antibiotics [8].

To address this critical concern, alternative antibacterial technologies have been actively pursued as potential substitutes for traditional antibiotic therapy to combat bacterial infections [9,10]. In contrast to antibiotics, nanomaterials offer multiple intrinsic passive antibacterial mechanisms [11] which target specific sites within bacterial cells, thereby reducing the risk of resistance development [12]. With the increasing integration and ongoing advancements of nanotechnology in the field of antimicrobials [13], nanomaterials have gradually been used as alternatives to antibiotics. The development and research of novel antibacterial agents based on nanomaterials have rapidly progressed over the past decade. These agents not only possess inherent advantages including broad-spectrum antimicrobial activity, high efficacy, and low toxicity, but also utilize multiple mechanisms to synergistically achieve antimicrobial effects, effectively mitigating the emergence of bacterial resistance [14]. For instance, nanomaterials can exert antimicrobial effects by physically interacting with bacterial cells, releasing metal ions, generating reactive oxygen species (ROS) directly, or enhancing ROS production indirectly. Additionally, the adjustability and optimization of the physicochemical properties of these nanomaterials make them highly versatile for integration into intelligent antibacterial strategies. This versatility allows them to effectively address the specific requirements of antibacterial fields.

Current research has demonstrated that nanomaterials can be employed based on their unique physicochemical properties to achieve inherent antibacterial effects by disrupting bacterial cell membranes or interfering with bacterial physiological activities [13]. Certain nanomaterials exhibit enzyme-like activities capable of disrupting bacterial biofilms, such as deoxyribonuclease-like DNase or peroxidase-like peroxidase (POD), thereby achieving bactericidal effects. Nanomaterial-based intelligent responsive platforms can be developed which can respond to infection microenvironments (e.g., pH, enzymes, hydrogen peroxide) and physical stimuli (e.g., light, ultrasound, magnetic fields) [15]. The development of nanoscale antimicrobial systems primarily entails drug loading and utilization of methods such as doping, composites, and surface modifications [16]. These strategies can be cleverly designed to achieve maximum antibacterial and minimal toxic side effects.

The emergence of antibiotic resistance has posed significant challenges in treating biofilm-related infections. Consequently, the pursuit of next-generation antimicrobial technologies should prioritize targeted therapies that bacteria cannot easily develop resistance against. Stimuli-responsive therapies present a promising alternative approach, as they have the ability to deliver precise and targeted treatments. This article provides an overview of recent research progress in the field of intelligent antimicrobial drug delivery systems that exhibit responsiveness to both endogenous and exogenous stimuli such as pH, enzymes, light, and ultrasound (Figure 1). These stimuli can activate nanostructures to achieve the specific release of drug molecules at certain times and locations, thereby avoiding the development of drug resistance and reducing cellular toxicity.

## 2. Formation of Bacterial Resistance and Antibacterial Strategies

Bacterial resistance refers to the ability of bacteria to withstand the effects of antimicrobial drugs, either by being insensitive to them or by gradually reducing their sensitivity through repeated use, ultimately leading to ineffective treatment [17]. Bacterial resistance is generally believed to be primarily achieved through various mechanisms, including the alteration of the cell wall structure to hinder the entry of antibiotics [18]; overexpressing inactivating enzymes that degrade antibiotics, rendering the antibiotics ineffective [9]; altering the structure and quantity of antibiotic target sites, diminishing the binding affinity between antibiotics and their targets [19]; and overexpressing efflux pumps to expel drugs, reducing intracellular drug concentrations to ineffective levels [20].

Moreover, bacteria have the ability to form biofilms, which are collective protective survival states (Figure 2). Biofilms serve as a defense mechanism, enabling bacteria to resist the penetration of antibiotics or immune substances. As a result, they exhibit reduced sensitivity to antibiotics and can induce bacterial resistance [5]. Studies have reported that the dosage of antibiotics needed to eradicate bacterial biofilms can be over 1000 times higher than that required to target planktonic bacteria. Furthermore, it is worth noting that currently, over 65% of human bacterial infectious diseases are associated with biofilms [21].

The formation of bacterial biofilms is a multifaceted and dynamic process that encompasses the initial contact and adhesion of planktonic bacteria to surfaces, followed by the release of autoinducers that induce the aggregation of similar bacteria [23]. Simultaneously, bacteria are embedded in the extracellular polymeric substances (EPSs) they secrete, which are composed of nucleic acids, proteins, polysaccharides, and other components. Gradually, they form bacterial communities and mature biofilms, which undergo cycles of detachment and reattachment [24]. Bacteria engage in intercellular communication through a process known as quorum sensing. This process entails the release of autoinducers, enabling them to regulate population density, control the generation of virulence factors, and facilitate the formation and maturation of biofilms [25]. The EPS within the biofilm acts as an effective physical and metabolic barrier, impeding the penetration of antibiotics. It restricts the mobility of bacteria within the biofilm, facilitating the transfer of genes responsible for antibiotic resistance between bacterial cells. The dense growth of bacteria within the biofilm triggers stress responses, resulting in the production of antibiotic-degrading enzymes [26].

The bacterial outer membrane or cell wall, being a crucial component in all bacterial cells, plays a vital role in maintaining cell shape, osmotic regulation, protection against mechanical stress, and defense against infection. Physical or mechanical damage to the bacterial outer membrane or cell wall can result in its dysfunction and the subsequent leakage of cytoplasmic components, ultimately leading to bacteriostatic and bactericidal effects [4]. This mechanism is widely acknowledged as one of the most common antibacterial pathways.

## 3. Characteristics and Mechanisms of Antibacterial Nanomaterials

Nanomaterials have proven to be successful in integrating potent adjuvant activity, high drug loading capacity, pathogen targeting, and site-specific drug release properties into a vesicle structure. Nanotechnology refers to an emerging interdisciplinary field that focuses on the study of atomic and molecular structure characteristics and their interactions at the nanoscale (typically ranging from 0.1 to 100 nm). It is an emerging interdisciplinary field and a prominent research area in modern materials science. Nanomaterials possess unique physical and chemical properties that are strongly influenced by their size. They can form three-dimensional, multi-component structures, enabling the delivery of hydrophobic molecules and overcoming various biological barriers, thereby selectively targeting specific sites in the living body [27,28,29].

Compared to existing antibiotics, nanomaterials offer complementary modes of action as antimicrobial agents. As shown in Figure 3, they can exert their unique effects, achieving antibacterial effects at cellular and in vivo levels with spatial and temporal dimensions [30]. The successful delivery of antimicrobial agents to the site of infection using nanocarriers can enhance the efficacy of therapeutic drugs in combating microbial infections [31]. Moreover, nanoparticle-based drug delivery systems can prolong the circulation time of drugs in the bloodstream, overcome nonspecific distribution at the site of infection, and enable targeted delivery.

Additionally, nanomaterial-based delivery systems can effectively address concerns regarding toxicity, and side effects that may arise from high-dose drug therapies. Nanomaterials with inherent therapeutic activity have demonstrated superiority over conventional antibiotics in overcoming antibiotic resistance and providing effective treatments [27]. The efficacy of nanoparticles varies depending on their concentration, shape, size, and surface atomic composition [32]. As the particle size decreases, the proportion of surface atoms to the total atoms in the bulk material increases, thereby enhancing the physical and chemical activities [33]. Consequently, compared to small-molecule antimicrobial agents, nanoparticles exhibit higher antimicrobial activity. ROS, including hydrogen peroxide (H_2_O_2_), superoxide anions (O^2−^), and hydroxyl radicals (·OH) are generated, which attack the membrane composed of polyunsaturated phospholipids, leading to the degradation of bacterial nucleic acids and impairment of bacterial cell functions [34].

The tunable physical and chemical properties of nanomaterials can effectively influence their antimicrobial activity and other pharmaceutical properties [35]. Moreover, the hydrophobicity, crystallinity, surface stability, surface roughness, electronic states, and curvature radius of nanomaterials differently influence protein binding, thereby modulating their biological activity [36].

The antimicrobial mechanisms of nanomaterials differ significantly from antibiotics that possess antimicrobial properties. Antibiotics achieve their antimicrobial effects by inhibiting cell wall synthesis, interfering with the expression of essential proteins, and disrupting DNA replication mechanisms [5]. With the distinctive physical and chemical properties, nanomaterials can utilize distinct antimicrobial pathways different from antibiotics to exhibit antimicrobial activity [37]. Nanomaterials can interact with bacterial cell membranes via specific physical and chemical interactions, leading to membrane disruption, leakage of cytoplasmic components, and binding to cellular components such as enzymes, ribosomes, and DNA, resulting in the impairment of normal cellular tissues [38]. This interaction can induce oxidative stress, disrupt electrolyte balance, and inhibit bacterial metabolism [39,40]. Apart from the direct antimicrobial effects that directly damage bacterial cells, certain nanomaterials with large surface areas can encapsulate bacterial cells, isolating them from their nutrient environment and leading to bacterial inactivation. Furthermore, some two-dimensional nanomaterials combined with other antimicrobial nanomaterials (such as metal and semiconductor nanoparticles, organic photosensitizers, and antimicrobial polymers) can demonstrate synergistic antimicrobial effects [41,42].

Therapies based on nanomaterials hold great promise in combating challenging bacterial infections, as they have the ability to circumvent acquired drug resistance mechanisms. Moreover, nanomaterials possess unique size and physical properties that enable them to target biofilms, effectively addressing recalcitrant infections. The design and synthesis of antibacterial nanomaterials with enhanced efficiency, stability, fouling resistance, biocompatibility, and recyclability represent a rapidly advancing field.

## 4. Stimuli-Responsive Antibacterial Nanomaterials

Stimuli-responsive intelligent antibacterial nanomaterials can be classified into endogenous stimuli-responsive and exogenous stimuli-responsive nanomaterials. Endogenous stimuli-responsive nanomaterials usually respond to the infected microenvironment, while exogenous stimuli-responsive nanomaterials typically respond to external physical stimuli. Nanomaterials responsive to bacterial metabolite stimuli are paving the way for the development of precision medicine and self-adaptive antibacterial systems. Bactericides can be exposed or precisely released on demand. Indeed, this approach holds significant promise in addressing the problem of overusing bactericides and mitigating the emergence of drug-resistant bacteria.

### 4.1. Endogenous Stimuli-Responsive Antibacterial Nanomaterials

The infected tissues and bacterial biofilms are known to possess a specific microenvironment that differs from normal tissues. This microenvironment is characterized by factors such as low pH, high ROS, upregulated enzymes, and more. The unique microenvironment in infected tissues and bacterial biofilms can be exploited to internally trigger specific properties of nanoparticles, including drug release, charge reversal, size change, and other functionalities. The stimuli-responsive behavior of nanoparticles offers a theoretical advantage, as it can be specifically activated upon reaching the infected site, providing a significant advantage in enhancing drug bioavailability.

The infection microenvironment refers to the localized environment in which the pathogen grows or persists within the infected host, such as bacterial biofilms. Significant differences have been observed between the infection site and normal tissue. The pH at the infection site is typically around 5 to 6.5, indicating a slightly acidic environment [43,44], whereas the pH in normal tissue is approximately 7.4. The infection site shows overexpression of specific enzymes, such as collagenase and hyaluronidase (Hydase) [45]. Additionally, elevated levels of H_2_O_2_, hydrogen sulfide (H_2_S), have been observed [46]. These characteristics have spurred research efforts to design intelligent antibacterial agents that can respond to stimuli in the infection microenvironment [30].

#### 4.1.1. pH-Responsive Antibacterial Nanomaterials

The microenvironment of bacterial infections is frequently characterized by acidity. Certain bacteria can lower the local pH of infected tissue through anaerobic fermentation induced by hypoxia. Furthermore, the host immune response can contribute to a decrease in local pH at the bacterial residence site through lactic acid production during phagocytosis. The acidic environment is considered an adverse factor that reduces the antimicrobial activity of certain antibiotics, such as erythromycin, sulfonamides, and other basic antibiotics. Based on nanomaterials or their surface properties, the pH-triggered response could be manifested as pH-triggered drug release and surface charge reversal. By leveraging the acidic environment specific to the site of bacterial infection, researchers can design targeted and controlled release systems or specific antibacterial effects in the treatment of bacterial infections [47,48,49,50,51]. The mechanism of pH-responsiveness can be categorized into two main groups: (1) protonation/deprotonation of amine groups and carboxyl groups and (2) cleavage of chemical bonds. Protonation of amine groups is a commonly employed approach to create pH-responsive nanoplatforms. As a result, numerous pH-responsive antibacterial nanomaterials have been reported, showing enhanced therapeutic efficacy in the treatment of bacterial infections.

In a study by Wu et al., pH-responsive photodynamic antibacterial nanoparticles were designed [52]. As shown in Figure 4a, to form the NP core (RB-PDA), the photosensitizer propylamine-functionalized Rose Bengal (RB-NH_2_), which generates singlet oxygen (^1^O_2_), was covalently conjugated with polydopamine (PDA). Moreover, Polymyxin B (PMB) binds to the outer membrane of Gram-negative bacteria, leading to the destabilization of bacterial outer membranes. Subsequently, gluconic acid (GA) was added via electrostatic interaction to form RB@PMB@GA NPs. RB@PMB@GA NPs maintain a negative charge under physiological pH conditions and demonstrate excellent biocompatibility. Upon exposure to an acidic infectious environment, the NPs undergo pH-sensitive electrostatic interactions, resulting in a conversion of the surface charge from negative to positive. The conversion of surface charge enables RB@PMB@GA NPs to efficiently bind to bacterial surfaces, thereby enhancing the photoinactivation efficacy against Gram-negative bacteria. Significantly, RB@PMB@GA NPs exhibit effective penetration into biofilms and eradication capabilities, particularly under acidic conditions. Moreover, RB@PMB@GA NPs demonstrate high efficacy in eradicating biofilm infections in vivo. By employing photosensitizers and appropriate wavelength excitation by light sources, ROS and primarily singlet oxygen (^1^O_2_) are generated, inducing oxidative damage to bacteria and offering a novel approach for bactericidal purposes [52]. Another study by Shi et al. proposed a pH-responsive/enzyme-cascade-reactive nanoplatform for antibacterial applications (Figure 4b) [53]. To create a multi-arm hydrophilic segment, the precursor molecule L-arginine (L-Arg) was modified with β-cyclodextrin (β-CD), resulting in the formation of β-CD-Arg. Furthermore, a pH-sensitive linear ferrocene-terminated acetal-modified maltoheptaose (Fc-AcMH) was utilized as the hydrophobic component in the system. The amphiphilic supramolecular structure is formed through the host–guest interaction between the β-CD cavity and ferrocene. Glucoamylase (GA) and glucose oxidase (GOx) were incorporated via electrostatic interactions to form Arg-CD-AcMH+GOx/GA. Under an acidic microenvironment, the acetal bond is cleaved, liberating maltoheptaose, which is subsequently hydrolyzed by GA to yield glucose. Gox facilitates the conversion of glucose to H_2_O_2_, leading to the oxidation of the guanidine moiety in L-Arg and the generation of NO. This process disrupts biofilms and exhibits bactericidal activity.

Further pH-responsive cage-like polymeric nanoparticles (CGNs) were synthesized by introducing acid-labile side chains into an amphiphilic diblock copolymer P(GEMADA-co-DMA)-b-PBMA and incorporating a hydrophobic photothermal agent [54]. Under physiological conditions and in blood circulation, the guanidine groups on the surface of CGNs are coupled with acid-labile groups, resulting in a negative charge and demonstrating excellent biocompatibility and blood compatibility. Upon passive targeting to the acidic environment of the infection site, the positively charged guanidine groups in CGNs are exposed, imparting CGNs with deep penetration and efficient bacterial adhesion. Subsequently, under NIR irradiation, the encapsulated CS in CGNs generates significant heat, leading to the efficient eradication of biofilms. (Figure 4c). In addition, Tan et al. proposed a design in which the chimeric peptide has the ability to self-assemble into nanofibers at pH 7.4 and undergo transformation into nanoparticles at pH 5.0 in the acidic microenvironment of biofilm-infected areas [55]. This design aims to effectively treat drug-resistant bacteria and infections associated with biofilms by achieving a reduction in size and an increase in charge, thereby enhancing penetration into bacterial biofilms. The mechanism of action is primarily based on membrane cleavage (Figure 4d).

In addition, Vijay et al. developed pH-responsive antibacterial nano-assemblies by encapsulating ferulic acid with a bioactive peptide amphiphile. These nano-assemblies exhibited shape transformation from nanospheres to nanofibers in the basic chronic wound environment, leading to enhanced antibacterial activity [56].

#### 4.1.2. Enzyme-Responsive Antibacterial Nanomaterials

Enzyme secretion is another characteristic of the site of bacterial infection. These enzymes create a unique microenvironment at the site of infection. Biomaterials such as HA, poly (ε-caprolactone), and polyphosphates can be degraded by these enzymes, leading to the release of encapsulated antibacterial agents. As a result, nanotargeted antibacterial therapy with such encapsulated antibacterial agents has garnered increasing attention.

Enzymes, as stimuli-responsive factors, are considered the most promising intelligent drug delivery strategy due to their high selectivity and efficiency under mild physiological conditions [57]. Resistant pathogens produce specific enzymes, such as penicillin amidase (PGA) and β-lactamase (βla), which play a significant role in the resistance to β-lactam antibiotics through catalytic hydrolysis (e.g., penicillin, cephalosporins, and carbapenems) [58,59]. Based on this, nanomaterials can respond specifically to enzymes due to their unique physicochemical properties. To mitigate the adverse effects of therapeutic drugs, intelligent nanomaterials can be modified with enzyme-labile linkages, enabling on-demand and responsive drug release upon enzyme stimulation [60,61,62,63].

In a study by Wu et al., they achieved successful synthesis of cage-like framework nanospheres (CFNSs) composed of bimetallic oxide Cu_1.5_Mn_1.5_O_4_, demonstrating enhanced triple enzyme activity, including oxidase, peroxidase, and glutathione peroxidase (GSH-Px) [64]. The precursors of CuMn-OH hollow microspheres were obtained through a gas-assisted soft template solvothermal method, followed by calcination at a certain temperature to rapidly obtain Cu_1.5_Mn_1.5_O_4_ CFNSs. This method offers a simple and rapid approach to synthesizing Cu_1.5_Mn_1.5_O_4_ CFNSs with well-dispersed and uniform sizes, along with mesoporous structures. Due to the increased surface area and unique structure of the CFNSs, more active edge sites are exposed, leading to the significant generation of ROS while effectively reducing intracellular GSH levels. Moreover, the material exhibits high antibacterial efficacy (Figure 5a).

To achieve controlled release of antimicrobial agents within nanoparticles or to mitigate nanoparticle toxicity, hyaluronic acid (HA) is commonly employed for nanoparticle encapsulation [68]. As shown in Figure 5b, Liu et al. used HA to modify mesoporous ruthenium (Ru) nanocarriers loaded with hydrogen peroxide prodrug (AA), followed by the coating of prefunctionalized molybdenum disulfide (MoS_2_) nanosheets with ciprofloxacin (CIP) [65]. This resulted in an enzyme-responsive antibacterial delivery system (AA@Ru@HA-MoS_2_). CIP was used for bacterial targeting, and when the system reached the vicinity of bacteria, it could be degraded by hydrolases secreted by the targeted bacteria, rapidly releasing AA. Subsequently, utilizing the peroxidase-like activity of MoS_2_, AA was converted into ·OH, efficiently killing the target bacteria. Moreover, mesoporous Ru NPs achieved photothermal synergy effect under near-infrared (NIR) light, leading to the eradication of bacterial biofilms and accelerated healing of local wound infections. Another study by Cheng et al. utilized Cu-MOF (CuBDC) based on Fenton-like catalytic activity and co-encapsulated glucose oxidase (GOx) and L-arginine (L-Arg) to generate L-Arg/GOx@CuBDC (Figure 5c) [66]. In response to glucose, a dual-cascade reaction takes place: initially, glucose oxidase (GOx) catalyzes the production of H_2_O_2_ from glucose, followed by an ROS cascade reaction in which H_2_O_2_ generates a significant quantity of ROS (such as ·OH and ·O^2−^) through Cu-MOF-mediated Fenton-like reactions. Simultaneously, in the cascade reaction of reactive nitrogen species (RNS), H_2_O_2_ oxidizes L-Arg to produce NO, and NO reacts with·O^2−^ to generate the highly toxic peroxynitrite anion (ONOO^−^). Bacterial inactivation exceeding 97% can be achieved at low concentrations of antibiotics (*E. coli*: 38 μg/mL; *S. aureus*: 3.8 μg/mL).

In another study, Li et al. successfully fabricated penicillin G amidase (PGA) and β-lactamase (βla)-responsive polymeric vesicles, as shown in Figure 5d [67]. The enzyme-mediated degradation processes and microstructural evolution of these vesicles were investigated. The polymeric vesicles loaded with antibiotics exhibited a self-immolative structural rearrangement and morphological transitions, leading to a sustained release of the antibiotics. This approach has achieved several significant advancements, such as improved structural stability, reduced side effects, and enhanced healing of burn wounds through the use of responsive vesicles loaded with antibiotics.

#### 4.1.3. High Levels of H_2_O_2_, H_2_S-Responsive Antibacterial Nanomaterials

Bacterial infection sites are frequently accompanied by elevated levels of H_2_O_2_ and H_2_S, which play a crucial role in the pathogenesis of infections [69]. Extensive research has demonstrated that virtually all bacteria possess the enzymatic machinery, including cystathionine γ-lyase (CSE), cystathionine β-synthase (CBS), or 3-mercaptopyruvate sulfurtransferase, enabling them to produce H_2_S as a protective response against oxidative stress. Inhibition of H_2_S production has emerged as a promising strategy for combating bacterial resistance [70,71,72,73,74,75].

Yang et al. developed a novel theranostic platform utilizing Cu_2_O nanoparticles (NPs). Upon in situ injection, the Cu_2_O NPs undergo a rapid reaction with endogenous H_2_S, resulting in the generation of Cu_9_S_8_ NPs, which can serve as NIR-II photoacoustic imaging agents for visualizing inflammatory tissues under 1060 nm irradiation [76]. Furthermore, H_2_S-activated Cu_9_S_8_ NPs exhibit excellent antibacterial properties in photothermal therapy (PTT) under 1060 nm irradiation. Additionally, Cu_2_O NPs effectively catalyze H_2_O_2_ to produce highly antibacterial ·OH through Fenton-like reactions, resulting in the disruption of bacterial cell membranes (Figure 6a).

Guo et al. proposed a spatially selective chemodynamic therapy strategy using bimetallic nanomaterials responsive to the microenvironment (pH and H_2_O_2_) [77]. Within the acidic microenvironment of the biofilm, CuFe_5_O_8_ nanocubes act as efficient catalysts, promoting the generation of abundant ·OH radicals, which effectively cleave extracellular DNA and disrupt the dense biofilm structure (Figure 6b). Moreover, in the extracellular environment outside the biofilm (characterized by higher pH and lower H_2_O_2_ concentration), CuFe_5_O_8_ nanocubes generate a controlled amount of ·OH radicals, leading to the reversal of the immunosuppressive microenvironment surrounding the biofilm. This process effectively activates local macrophages and facilitates the clearance of damaged biofilms and floating bacteria. The endogenous stimuli-responsive antibacterial nanomaterials are compiled as follows (Table 1), which only presents a portion of examples, and there are many other fascinating works that have not been included.

### 4.2. Exogenous Stimuli-Responsive Antibacterial Nanomaterials

Exogenous stimuli-responsive nanoparticles can be activated by external stimuli, such as light, magnetic fields, electric fields, ultrasound, and more. Given the ease of controlling these external stimuli, exogenous stimuli-responsive nanoplatforms hold great promise in achieving spatiotemporally controlled drug delivery. Additionally, the design of responsive antibacterial surfaces requires careful consideration of stimuli-responsive materials.

Despite the fabrication of various endogenous stimuli-responsive nanoplatforms, achieving accurate control over drug release remains challenging due to the complex physiological environment and significant individual differences. In this context, the development of exogenous stimuli-responsive nanoplatforms becomes advantageous, as external stimuli can be readily controlled.

Certain nanomaterials possess exceptional photothermal, photodynamic, sono-thermal, and sonodynamic effects when stimulated by exogenous energy sources, including light, ultrasound (US), magnetic fields, and microwaves (MW). They demonstrate reduced susceptibility to bacterial resistance and hold significant potential in the field of antimicrobial applications [78]. Consequently, the development of intelligent nanoscale antimicrobial agents that utilize the distinctive properties of nanomaterials to respond to external stimuli, including light, sound, and magnetic fields, has emerged as a highly active area of research [79].

#### 4.2.1. Photo-Responsive Antibacterial Nanomaterials

Photo-responsive antibacterial nanomaterials typically incorporate chromophores capable of converting light stimuli into chemical or energy outputs; whether based on their intrinsic optical properties or in conjunction with photosensitizers and photothermal agents, they play a critical role in light-mediated therapeutic strategies [80]. These strategies encompass a range of techniques, including photothermal therapy (PTT) and photodynamic therapy (PDT) [81]. PTT is a thermal-based therapy technique that utilizes nanomaterials with high photothermal conversion efficiency to convert light energy into heat energy when exposed to an external light source. High temperatures generated by photothermal effects (>42 °C) can induce bacterial cell apoptosis, disrupt cell membranes, damage the cell cytoskeleton, and inhibit DNA synthesis. PDT has emerged as a promising approach to effectively eradicate pathogenic bacteria by utilizing a photosensitizer and laser irradiation. PDT oxidizes biomolecules and inflicts irreversible damage through the generation of ROS [81]. A notable advantage of PDT over conventional antibiotic-based therapies is its capacity for repeated treatment without inducing undesirable drug resistance [82,83,84,85]. PTT and PDT have become promising antibacterial methods due to their low invasiveness, low toxicity, and avoidance of drug-resistant bacteria.

As shown in Figure 7a, Li et al. developed methylene blue (MB) and lysozyme (LYZ)_-_loaded upconversion nanoparticles (UCNPs, which are nanoscale particles capable of converting lower-energy light into higher-energy light through nonlinear optical processes) [86]. They then assembled HA and poly-ε-lysine (ε-PL) layer by layer on the outer surface, resulting in UCMB-LYZ-HP. In the infection microenvironment, HA was degraded by hydrolases, leading to the release of LYZ. Under 980 nm irradiation, MB achieved PDT to effectively eliminate *MRSA* (with a decrease in *MRSA* viability > 5 log10).

When photothermal therapy (PTT) is applied alone, it often requires higher temperatures to achieve a substantial antibacterial effect. However, such elevated temperatures may also cause damage to normal tissues and trigger new inflammation. To mitigate these concerns, researchers have explored the combination of PTT with other antibacterial methods, which has shown promising results in reducing side effects on normal cells while enhancing the overall therapeutic effect. Numerous reports have confirmed that integrating PTT with other techniques, such as photodynamic therapy (PDT) or additional antibacterial approaches, can effectively minimize adverse effects on normal cells and improve the treatment’s efficacy [89]. As shown in Figure 7b, Xu et al. synthesized a multifunctional composite hydrogel (PDA/Cu-CS) using polydopamine (PDA) and copper-doped calcium silicate ceramic (Cu-CS) as the main components [87]. By leveraging the unique “hot ions effect” (some ions in the plasma have higher energy than ordinary ions, and these high-energy ions damage the cell wall and membrane of microorganisms, resulting in cell death, so as to achieve bactericidal effect) generated through the photothermal effect of the composite hydrogel, the heating of Cu ions resulted in a highly efficient, rapid, and long-lasting inhibition of *MRSA* and *Escherichia coli.* Additionally, the hydrogel exhibited remarkable bioactivity in promoting angiogenesis.

Dai et al. prepared a water-soluble galactose-decorated cationic 4,4-difluoro-4-bora-3a,4a-diaza-s-indacene (BODIPY)-based PDT agent (P(ATA-C4)-r-GAL-I2) for selectively eliminating *P. aeruginosa*, *S. aureus*, and bacterial biofilm (Figure 7c) [88]. Under visible light irradiation, P(ATA-C4)-r-GAL-I2 can efficiently attach to the bacterial surface through electrostatic interactions and subsequently induce the generation of a significant quantity of ROS, which irreversibly disrupt the bacterial membrane to kill the bacteria. The photosensitizer exhibits a remarkable ability to inhibit and eliminate over 70% of bacterial biofilms even at a very low concentration (22 μg/mL), all without inducing bacterial resistance.

Immobilization of AgNPs on a polymer matrix to form nanogels has demonstrated good antibacterial performance [90,91]. Similarly, gold nanorods decorated with phospholipid-polyethylene glycol and loaded into poloxamer 407 hydrogel (DSPE-AuNR) exhibited photothermal-induced antibacterial activity against both Pseudomonas aeruginosa planktonic and biofilm cultures [92]. Overcoming antimicrobial resistance using electrospinning and 3D printing of polymeric-based nanomaterials is also a current research hotspot [93].

#### 4.2.2. Thermally Responsive Antibacterial Nanomaterials

Thermally responsive nanomaterials are a distinct type of material different from light-thermal nanomaterials, which are considered another type of “smart” nanomaterial because they are able to change their physical properties (e.g., shape, size, etc.) in response to mild temperature changes, triggering the controlled release of drug [94,95]. Thermo-responsive polymers experience a phase transition at their critical solution temperature, resulting from the disturbance of intra- and intermolecular interactions. This behavior leads the polymer to either expand or collapse within the aqueous solvent. Polymers with a lower critical solution temperature (LCST) demonstrate phase separation, such as precipitation, when the temperature exceeds a specific threshold. Conversely, polymers with an upper critical solution temperature display phase separation below a specific temperature. In hydrogel systems, where the swelling increases or decreases around this transition temperature, researchers sometimes refer to this swelling transition as the volume phase transition temperature.

Thermo-responsive materials can be classified into two categories based on their solubility in water. Thermo-responsive materials exhibiting transition temperatures close to body temperature are widely employed in the biomedical field. Among these materials, poly(n-isopropylacrylamide) (pNIPAM) is the most commonly utilized thermo-responsive polymer. It can undergo a reversible transformation from a hydrophilic coil state to a hydrophobic globule state near 32 °C, with this transition being modulated by varying the polymer concentration and incorporating surfactants and copolymers. [96]. pNIPAM is also commonly crosslinked to form hydrogels with reversible thermal shrinkage and expansion properties. Its ease of functionalization makes pNIPAM a versatile candidate, often utilized as a hydrophilic segment in the synthesis of thermo-responsive amphiphilic block copolymers. Therefore, pNIPAM is an excellent polymer for developing materials with controllable antimicrobial properties [97,98].

As shown in Figure 8a, Zhan et al. proposed a novel approach to create a smart antimicrobial surface by immobilizing antimicrobial peptides (AMPs) on the surface [99]. The process involved the formation of a polydopamine film on the substrates, followed by the conjugation of pNIPAM to the film using atom transfer radical polymerization (ATRP). Subsequently, AMPs were conjugated to the pNIPAM units on the pNIPAM-modified surface through a click chemistry reaction. The temperature-sensitive nature of pNIPAM allowed for the exposure of the AMPs motif to the external environment at room temperature (25 °C), resulting in higher antimicrobial activity compared to body temperature (37 °C). Notably, this smart behavior was accompanied by increased biocompatibility of the surface at body temperature, surpassing the unmodified or individually modified substrates with AMPs or pNIPAM alone. This study suggests that the implants exhibit antibacterial properties at room temperature and can be safely used during surgery, leading to infection-free implantation. It represents a promising strategy for exploring the antimicrobial potential of pNIPAM-based nanomaterials while ensuring biocompatibility at physiological temperatures.

In another study, Yan et al. developed a sprayable hydrogel that undergoes in situ formation. The hydrogel is composed of a copolymer consisting of poly(N-isopropylacrylamide_166_-co-n-butyl acrylate_9_)-poly(ethylene glycol)-poly(N-isopropylacrylamide_166_-co-n-butyl acrylate_9_), which is referred to as PEP [100]. It also incorporates silver-nanoparticles-decorated reduced graphene oxide nanosheets, denoted as AG. This thermo-responsive hydrogel exhibits interesting sol–gel transition behavior at low temperatures, making it suitable for stable wound dressing applications, which is attributed to the inorganic/polymeric dual network and abundant coordination interactions between Ag@rGO nanosheets and pNIPAM (Figure 8b).

A series of temperature-responsive antibacterial nanomaterials, consisting of polymer-grafted doped silver nanoparticles, have been successfully prepared [101,102,103,104,105,106]. These nanocomposites demonstrate excellent biocompatibility and exhibit morphological and size changes in response to temperature variations. Moreover, they can release drugs on demand. The incorporation of silver nanoparticles endows the nanomaterials with outstanding antibacterial properties, effectively inhibiting the proliferation of various bacteria. As a result, they hold great potential for further development.

#### 4.2.3. Ultrasound/Microwave-Responsive Antibacterial Nanomaterials

Ultrasonic- or microwave-responsive nanomaterials can usually generate ROS or heat under ultrasonic and microwave stimulation, which are highly cytotoxic to various drug-resistant bacteria [107,108,109,110,111,112]. This shows great potential in the treatment of deep infections due to its advantages of non-invasiveness, excellent tissue penetration, and limited ultrasound site irradiation, thus producing antibacterial effects [113,114]. In a study by Yang et al., they reported ultrasonic-responsive antibacterial mesoporous silica-based nanomaterials. They utilized mesoporous silica-coated titanium dioxide nanoparticles with thiolated surface functionalization (TiO_2_@MS-SH) as crosslinkers [115]. The nanoparticles reacted with norbornene-functionalized dextran (Nor-Dex) through ultrasound-triggered thiol-norbornene reactions, leading to the formation of TiO_2_@MS-SH/Nor-Dex nanocomposite hydrogels. The TiO_2_@MS-SH/Nor-Dex nanocomposite hydrogel exhibited antibacterial activity by functioning as an ROS agent under ultrasonic irradiation.

As shown in Figure 9a, Sun et al. proposed the use of ultrasound-driven micro/nanomotors to enhance sono-dynamic antibacterial efficiency [116]. Specifically, they presented meso-tetra(4-sulfonatophenyl) porphyrin (TPPS)-modified gold nanorod (Au NR) motors as an illustrative example. The TPPS@Au nanomotors have impressive features, which can produce ROS for bacterial inactivation while performing multimodal motion (straight movement, rotation, vibration, and circular movement) via actuation using low-frequency ultrasound in different biological fluids such as water and bacterial suspensions.

Another study by Wei et al. designed a microwave-responsive system with Na^+^ insertion into Prussian blue (PB), showcasing its effectiveness in treating deep bacterial infectious osteomyelitis (Figure 9b) [114]. The application of microwaves stimulated PB through dielectric loss and alteration of the spin state of iron ions, resulting in heat generation and weakening of the Fe II-(CN) and Fe III-(NC) bonds in physiological saline solution. Simultaneously, the insertion of Na^+^ into PB caused the bond energy to become irreversible, thereby accelerating the release of Fe^2+^ and Fe^3+^ ions from PB. Under microwave treatment, the released Fe^2+^ and Fe^3+^ ions effectively penetrated the bacterial membrane, which exhibited reduced permeability. This penetration of ions into the bacterial cells resulted in the death of bacteria through the synergistic effects of microwaves, microwave-induced thermal effects, and the Fe^2+^/Fe^3+^-induced Fenton-like reaction, coupled with the consumption of GSH.

#### 4.2.4. Magnetic-Responsive Antibacterial Nanomaterials

Magnetic-responsive nanomaterials can be activated by magnetic fields, leading to the development of magnetically activated antimicrobial technologies. Compared to light-based approaches, magnetic activation offers the advantage of deep tissue penetration and activation due to the transparency of human tissue towards magnetic fields. Two main mechanisms have been explored for magnetically activated antimicrobial behavior: (1) magnetic hyperthermia, which involves localized temperature changes induced by magnetic activation, and (2) magnetophysical action, where nanomaterials exhibit antimicrobial behavior driven by kinetic forces in response to an applied magnetic field [117,118,119].

Elbourne et al. developed magnetic-responsive gallium-based liquid metal droplets (GLM-Fe) as antibacterial materials capable of physically damaging, disintegrating, and killing pathogens within mature biofilms of both *Pseudomonas aeruginosa* and *Staphylococcus aureus* [120]. The application of a magnetic field induces spinning and shape transformation of the GLM-Fe particles, resulting in the exertion of physical forces on the bacteria. This process disrupts the dense biofilm matrix and simultaneously leads to the lysis of the bacterial cells. The high bactericidal efficacy is achieved through the combined effects of shape transformation and magnetic activation (Figure 10a).

Liu et al. utilized MXene, Au, and polydopamine as raw materials with excellent photothermal properties to synthesize antibacterial agents (MXene@Fe_3_O_4_/Au/PDA) with excellent photothermal magnetic coupling properties (Figure 10b) [121]. The antibacterial mechanism is primarily attributed to the direct transfer of heat generated by the photothermal effect of the nanosheet to the cell membrane. Additionally, the edges of the nanosheets acts as a magnetic “hot nanoblade,” which leads to the shrinkage, deformation, and even rupture of the cytomembrane. This process contributes to the effective destruction of bacterial cells.

The exogenous stimuli-responsive antibacterial nanomaterials are compiled as follows (Table 2), which only presents a portion of examples, and there are many other fascinating works that have not been included.

The field of stimuli-responsive intelligent nanomaterials is constantly evolving and optimizing. The examples provided here are just the tip of the iceberg, aimed at sparking interest and encouraging further exploration in this exciting area of research.

## 5. Conclusions and Outlook

The current challenges in antimicrobial research encompass antimicrobial resistance, biocompatibility and safety concerns, and the persistence and stability of antimicrobial materials. Drug-resistant bacteria have diminished the effectiveness of traditional antibiotics, highlighting the need for alternative antimicrobial strategies.

Extensively investigated stimuli-responsive antibacterial nanomaterials capable of responding to both endogenous and exogenous stimuli demonstrate significant potential in combating bacterial resistance.

Stimuli-responsive nanocarriers offer significant advantages over conventional nanocarriers, as they enable on-demand drug release, improved drug accumulation, and extended retention at infected sites. These benefits stem from their sophisticated nanoplatform design, which enhances interactions with bacterial cells. As a result, these nanocarriers are highly effective in targeted drug delivery for combating infections and achieving better therapeutic outcomes. Infected sites and bacterial biofilms possess a distinct microenvironment, differing from normal tissues. This environment is characterized by factors such as low pH, overexpression of specific enzymes, high concentrations of H_2_O_2_, and secretion of specific toxins. Exploiting this unique microenvironment, we can design endogenous stimuli-responsive nanoplatforms that enable targeted release of bactericidal agents in response to the infected conditions. This approach enhances the bioavailability of antibiotics and improves their efficacy, particularly for combating bacterial infections. The pH levels at infected sites vary across different body parts, and infections caused by different bacterial strains may result in diverse microenvironments. Additionally, it is essential to take into account that the degree of infection plays a significant role in shaping the microenvironment. This consideration becomes crucial while designing endogenous stimuli-responsive nanoplatforms. Exogenous stimuli-responsive nanoplatforms are equally important and have captured significant attention from researchers, especially in the realm of antibacterial applications. Light-responsive nanocarriers offer immense potential for achieving precise spatiotemporal drug release and pulsed delivery, presenting promising prospects for practical applications. The design of light-responsive nanomaterials stands at the forefront of current research, and the integration of composite light-responsive materials greatly enhances the application capabilities of light-responsive nanoparticles. Combined approaches, such as photo-responsive/antibiotic or photo-responsive/thermo-responsive combination therapies, exhibit substantial potential in eradicating bacterial infections. Their greater penetration depths compared to light enable these stimuli to act as powerful release triggers for combating internal bacterial infections. The considerable potential of utilizing magnetism and ultrasound as exogenous stimuli for nanoplatforms calls for further research and investigation.

Despite the promising prospects of intelligent stimuli-responsive nanomaterials for antimicrobial applications, several challenges and issues need to be addressed during their development and widespread implementation:

(1) Safety Assessment: Comprehensive safety evaluations, including biocompatibility, toxicity, and metabolic pathways, are crucial for smart nanomaterials as novel agents. Ensuring their safety for human and environmental use is paramount, particularly for clinical applications.

(2) Lack of Standardization: The lack of uniform evaluation criteria and testing methods for smart nanomaterials results in incomparable outcomes and difficulties in generalizing their applications across different research institutions and countries.

(3) Antimicrobial Resistance: Although smart nanomaterials enable precise targeting of infection sites, rapid bacterial adaptability may lead to reduced antimicrobial efficacy and the emergence of resistance. Measures to counter antimicrobial resistance need to be implemented.

(4) Controlled Drug Release: Ensuring controlled drug release from smart nanomaterials is crucial to prevent either excessive or insufficient drug delivery, which could compromise treatment effectiveness.

(5) Large-Scale Production Costs: Complex manufacturing processes of smart nanomaterials might result in higher production costs. Further research and process optimization are required to lower costs and facilitate their mass production for clinical and industrial applications.

(6) Long-Term Stability: Ensuring the long-term stability of smart nanomaterials is vital, ensuring they retain their activity and do not undergo unpredictable changes during storage and use.

The future clinical prospects of smart responsive antimicrobial nanomaterials are highly promising. Intelligent responsiveness allows for personalized treatments tailored to individual needs and specific diseases, enhancing therapeutic efficacy while minimizing side effects. The application of smart responsive materials in medical devices, hospital surfaces, and wound management can effectively control infections and promote wound healing. We believe that smart responsive nano-antibacterial materials have a good application prospect in the future, and their biosafety, cost, yield and antibacterial effect will be recognized. In the future, the integration of nanomedicine with cutting-edge technologies such as artificial intelligence can be explored to construct more precise and intelligent disease diagnosis and treatment systems, opening up new directions for the development of biomedical science.

## Figures and Tables

**Figure 1 pharmaceutics-15-02113-f001:**
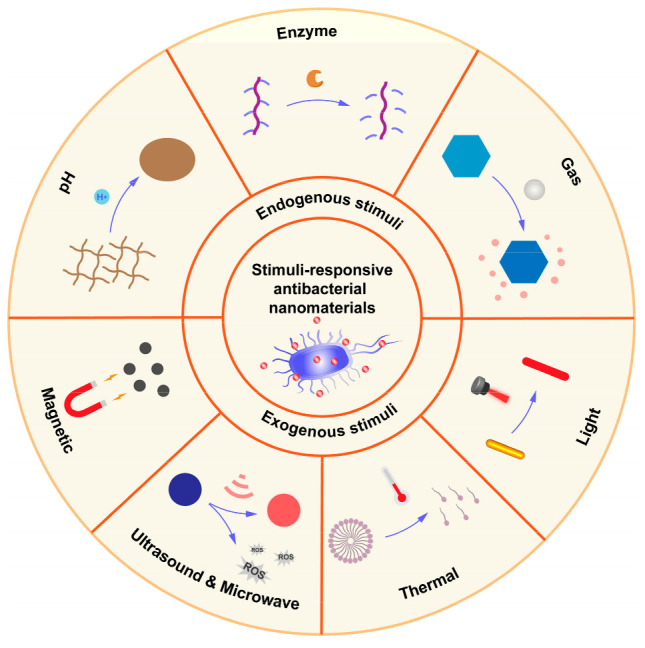
Endogenous and exogenous responsive-stimuli antibacterial nanomaterials.

**Figure 2 pharmaceutics-15-02113-f002:**
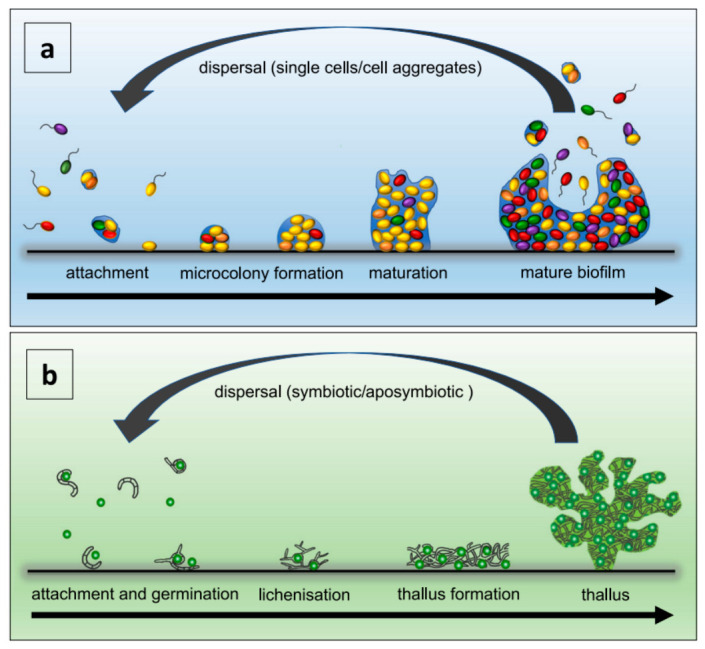
Biofilm development (**a**) and vegetative reproduction of lichens (**b**) are depicted. Panel a illustrates the primary stages of biofilm formation. Panel b showcases the key steps involved in lichen’s vegetative reproduction [22], Copyright 2021 Nature portfolio.

**Figure 3 pharmaceutics-15-02113-f003:**
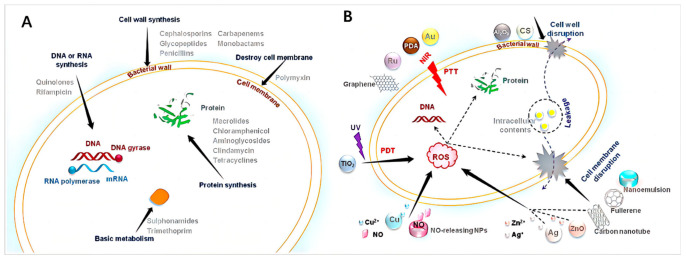
Current antibacterial strategies: (**A**) antibacterial targets of antibiotics and (**B**) various antibacterial mechanisms of antibacterial nanoparticles [30], Copyright 2022 Elsevier.

**Figure 4 pharmaceutics-15-02113-f004:**
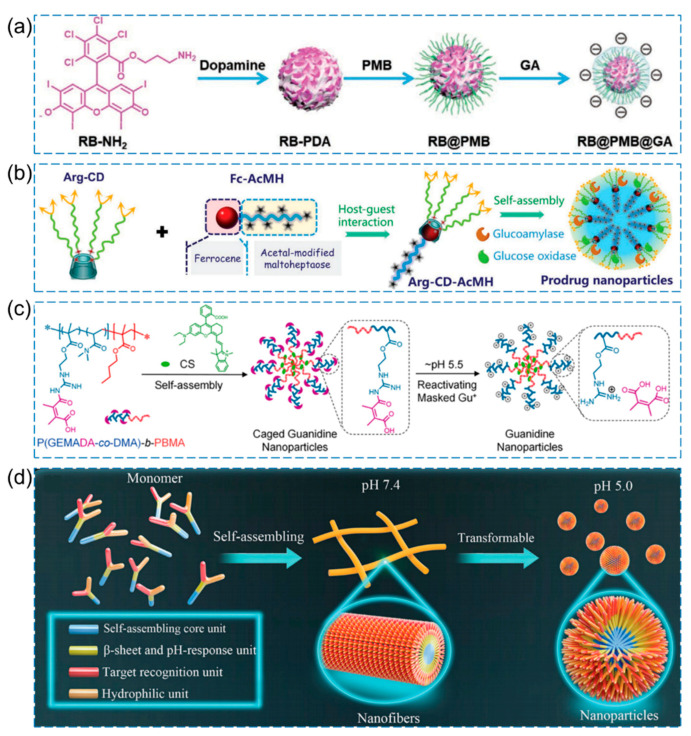
Examples of pH-responsive antibacterial nanomaterials. (**a**) Schematic illustration of the construction of RB@PMB@GA NPs [52], Copyright 2021 Wiley-VCH. (**b**) Schematic illustration for the synthesis of Arg-CD-AcMH+Gox/GA [53], Copyright 2022 Wiley-VCH. (**c**) Schematic illustration for the fabrication of P(GEMADA-co-DMA)-b-PBMA [54], Copyright 2020 American Chemical Society. (**d**) Schematic diagram illustrating the pH-responsive peptide nano-assemblies [55], Copyright 2023 Wiley-VCH.

**Figure 5 pharmaceutics-15-02113-f005:**
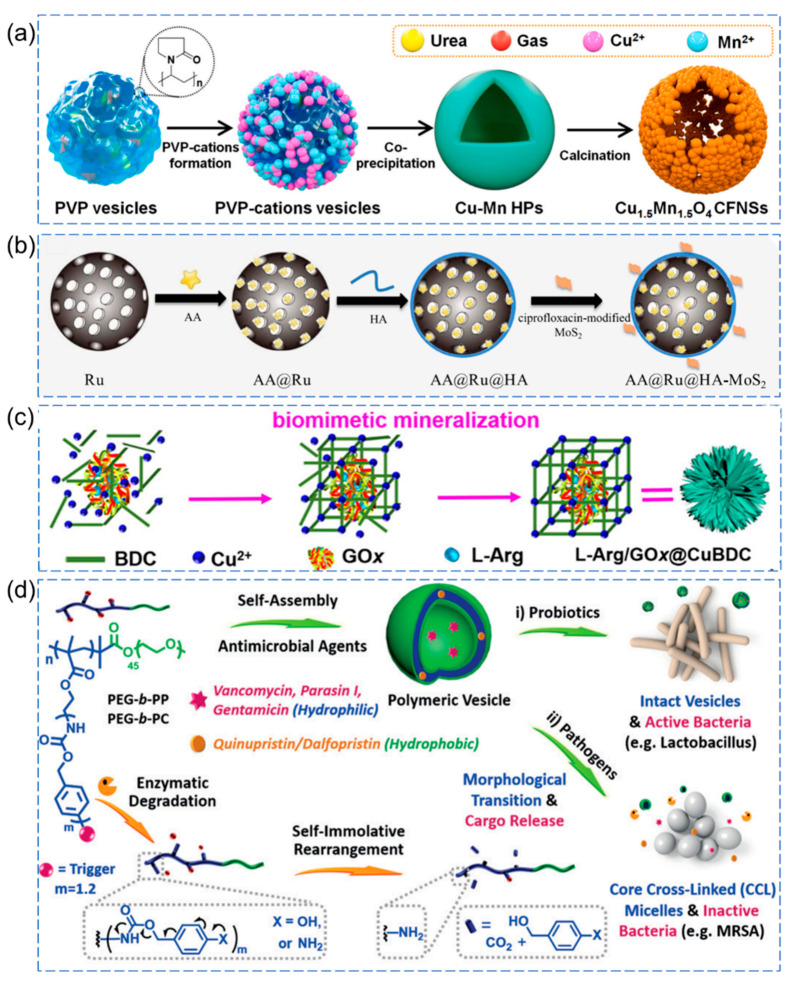
Examples of enzyme-responsive antibacterial nanomaterials. (**a**) Schematic illustration of preparation of Cu_1.5_Mn_1.5_O_4_ CFNSs [64], Copyright 2022 Elsevier. (**b**) Schematic illustration of enzyme-responsive antibacterial delivery system AA@Ru@HA-MoS_2_ [65], Copyright 2019 Elsevier. (**c**) Schematic illustration of preparation of L-Arg/GOx@CuBDC [66], Copyright 2020 American Chemical Society. (**d**) Schematic illustration of enzyme-responsive polymeric vesicles for bacterial strain selective delivery of antibiotics [67], Copyright 2016 Wiley-VCH.

**Figure 6 pharmaceutics-15-02113-f006:**
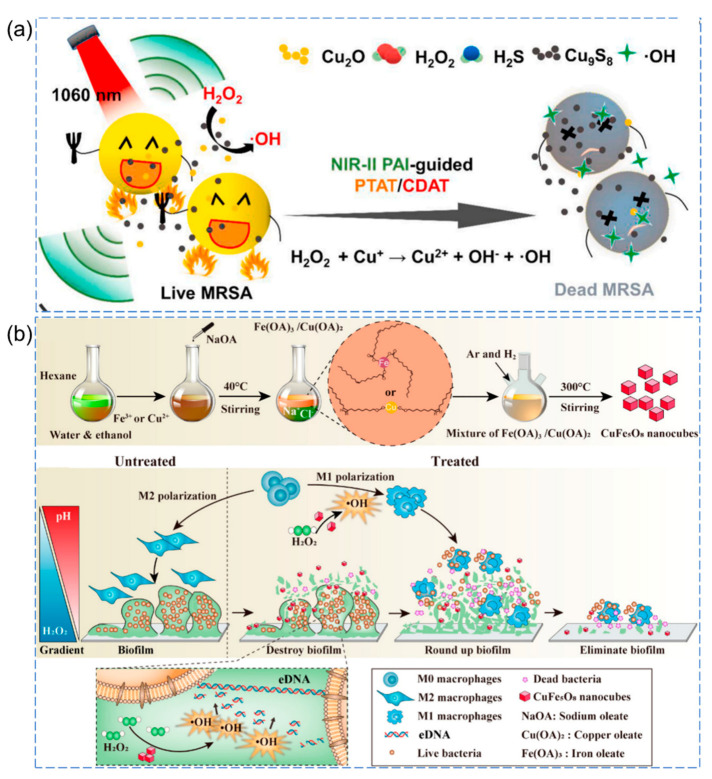
(**a**) Schematic representation of Cu_2_O nanoparticles activated by the infection microenvironment for NIR-II photoacoustic imaging-guided photothermal synergistic therapy [76], Copyright 2021 Elsevier. (**b**) Schematic illustration of basic synthetic processes and the proposed antibiofilm and immunomodulatory mechanisms of CuFe_5_O_8_ nanocubes [77], Copyright 2020, American Chemical Society.

**Figure 7 pharmaceutics-15-02113-f007:**
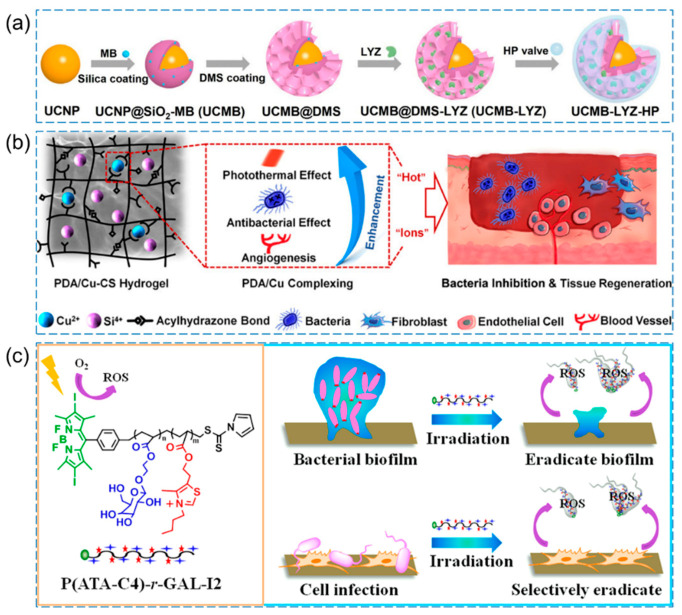
(**a**) Schematic diagram depicting the fabrication process of antibacterial nanohybrid UCMB-LYZ-HP [86], Copyright 2021 Wiley-VCH. (**b**) Design and application of the PDA/Cu-CS composite hydrogel [87], Copyright 2021 American Chemical Society. (**c**) Presentation of P(ATA-C4)-r-GAL-I2 for selective eradication of bacteria and bacterial biofilm [88], Copyright 2021 American Chemical Society.

**Figure 8 pharmaceutics-15-02113-f008:**
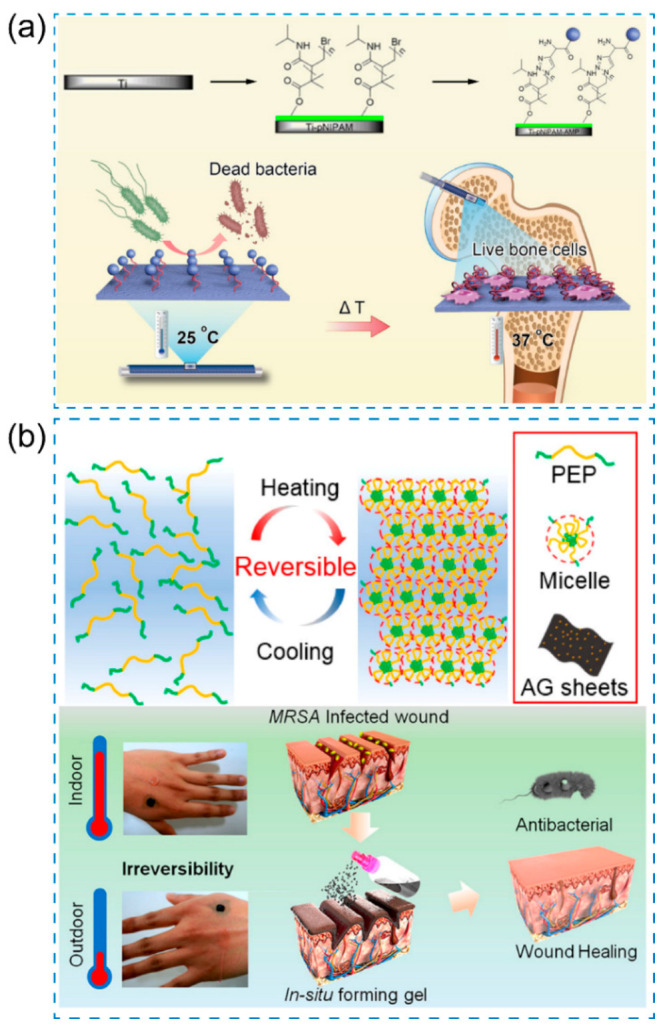
(**a**) Schematic illustration of pNIPAM-AMP dual modified bone implants for antibacterial treatment [99], Copyright 2018 American Chemical Society. (**b**) Schematic illustration of the thermo-responsive Ag@rGO hydrogel for antibacterial treatment [100], Copyright 2019 American Chemical Society.

**Figure 9 pharmaceutics-15-02113-f009:**
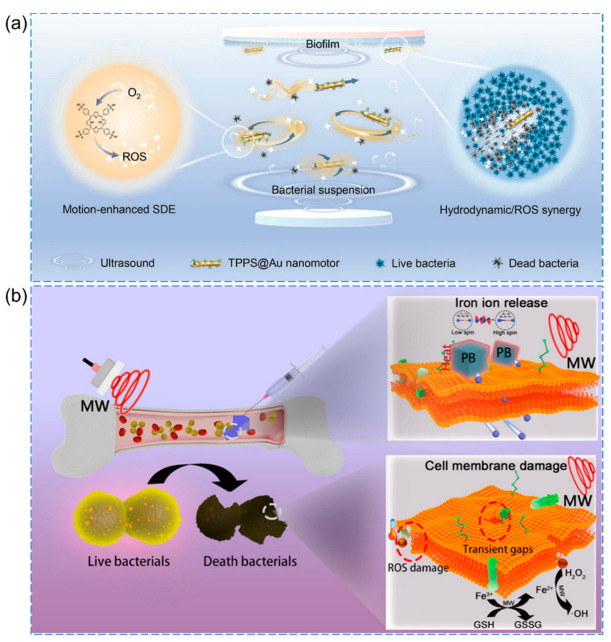
(**a**) Schematic illustration of an ultrasound-propelled sonodynamic nanomotor for enhanced bacterial inactivation [116], Copyright 2023 Wiley-VCH. (**b**) Schematic illustration of Na^+^ inserted metal–organic framework for rapid therapy of infected bacteria [114], Copyright 2021 Elsevier.

**Figure 10 pharmaceutics-15-02113-f010:**
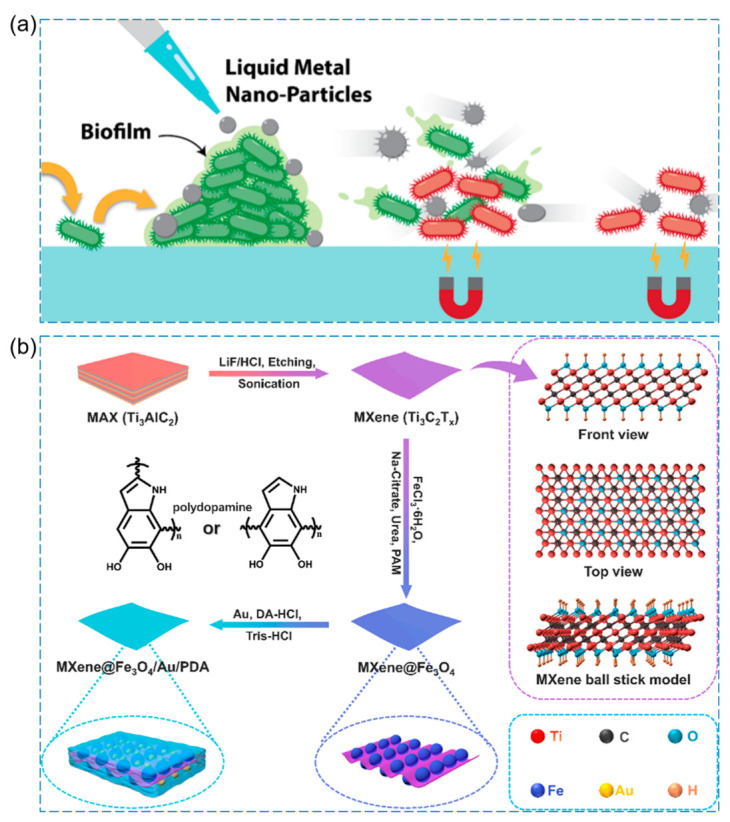
(**a**) Antibacterial schematic diagram of magneto-responsive gallium-based liquid metal droplets (GLM-Fe) [120], Copyright 2020 American Chemical Society. (**b**) Schematic illustration for preparation of the MXene@Fe_3_O_4_/Au/PDA nanosheet. Insert: MXene ball stick model [121], Copyright 2022 Elsevier.

**Table 1 pharmaceutics-15-02113-t001:** Examples of endogenous stimuli-responsive antibacterial nanomaterials.

Antibacterial Nanomaterials	Triggers	Nanocarriers	Bactericidal Moieties	Bacteria/Biofilm	Ref.
AZM-DA NPs	pH	multi-segment graft copolymer	azithromycin (AZM)	in vitro: *P. aeruginosa* (AZM equivalent: 8 μg/mL)in vivo: *P. aeruginosa* biofilm:(AZM equivalent: 25 mg/kg)	[43]
Ag nanoparticle clusters (AgNCs)	pH	functional polymers	release Ag^+^	MIC: *MRSA* (4 µg/ mL) and *E. coli* (8 µg/ mL)MBC: *MRSA* (32 µg /mL) and *E. coli* (32 µg /mL)	[47]
PPEGMA-AuNRs	pH	polymethacrylate (PCB)polymethacrylate with pendant mPEG (PPEGMA)	pH-induced surface charge-transformable, biofilm elimination	MIC: *E. coli* and *S. aureus* (31.25 μg Au/mL) *MRSA* and EBSL *E. coli* (125 μg/mL)	[48]
ferulic acid-encapsulated nanoparticles (FA-NPs).	pH	amphiphile peptide	ferulic acid	*MIC: E. coli* (750 μg/mL), *S. aureus* (900 μg/mL)	[56]
AA@GS@HA-MNPs	enzyme (hyaluronidase)	graphene-mesoporous silica nanosheet@hyaluronic acid-magnetic nanoparticles	ascorbic acid (AA)vancomycin	in vitro: *E. coli* and *S. aureus* (AA equivalent: 4 mg/mL)in vivo: *S. aureus* Biofilm (AA equivalent: 1 mg/mL)	[45]
Enzyme-responsive polymeric vesicles	enzyme: penicillin G amidase (PGA) and β-lactamase (Bla)	polymeric vesicles	structural rearrangement and morphological transitions, release PGA and Bla	*MRSA, B. longum, L. acidophilus*, and *E. faecalis* (1.0 μg/mL)	[67]
Ag-mesoporous silica nanoparticles (Ag-MONs)	GSH	mesoporous organosilica nanoparticles (MONs)	Ag NPssilver nitrate	*E. coli and S. aureus.* (1.28 μg/mL)	[46]

**Table 2 pharmaceutics-15-02113-t002:** Examples of exogenous stimuli-responsive antibacterial nanomaterials.

Antibacterial Nanomaterials	Triggers	Nanocarriers	Bactericidal Moieties	Bacteria/Biofilm	Ref.
nanogel containing silver nanoparticles (AgNPs)	light	polycaprolactone (PCL) nanofibers mats	release Ag+: disrupts ATP production and DNA replicationROS damage cell membranes	*S. aureus* and *E. coli* (57.6 μg/mL)	[88]
DSPE-AuNR	light	Polymeric Hydrogel	Photothermal-induced antibacterial activity	*P. aeruginosa* biofilm (0.25–0.03 nM)	[90]
ZnTCPP@ZnO	ultrasound	MOF	ROS	*Propionibacterium acnes*	[107]
BM2-LVFX-NPs	ultrasound	levofloxacin-loaded PLGA-PEG nanoparticles with BM2 aptamer	ROS	*Bacillus Calmette-Guérin* bacteria	[110]
RBC-HNTM-Pt@Au	ultrasound	Au NRs-actuated single-atom-doped porphyrin MOF (HNTM-Pt@Au)	ROS, dynamically neutralize the secreted toxins	*MRSA*	[111]
HNTM/Nb_2_C	ultrasound	Nb2C nanosheet-decorated porphyrin MOF hollow nanotubes (HNTM/Nb_2_C)	ROS (the rapid charge transfer and suppressed recombination of electron–hole pairs)	*MRSA*	[113]
Na+ inserted PB system	microwave	MOF, Prussian blue (PB)	Fenton reaction and thermal effects	*E. coli* and *S. aureus*	[114]
TiO_2_@MS-SH/Nor-Dex nanocomposite hydrogels	ultrasound	mesoporous silica-coated TiO_2_ nanoparticles with thiolated surface functionalization (TiO_2_@MS-SH), norbornene-functionalized dextran (Nor-Dex)	ROS	*E. coli* and *S. aureus*	[115]
NCJC/Fe_3_O_4_/Ag	magnetic field	nanocrystalline jute cellulose (NCJC) particles	Ag^+^	*S. aureus*, *E. coli*, *S. dysenteriae*, *S. boydii*, *Shigella boydii* (5 μg/mL)	[117]
TA-CFO/PVA	magnetic field	tannin (TA), cobalt ferrite nanoparticles (CFO NPs), polyvinyl alcohol (PVA) matrix	TA, Co^2+^	*E. coli* and *S. aureus*	[118]
GLM-Fe	magnetic field	Galinstan-based liquid-metal microparticles and nanoparticles (GLM-Fe)	magnetic field induces the GLM-Fe particles to spin, shape-transform, and impart physical forces to the bacteria	*P. aeruginosa* and *S. aureus* (100 μg/mL)	[120]
MXene@Fe_3_O_4_/Au/PDA nanosheets	magnetic field/light	polydopamine	PTT, nanosheet cuts the cytomembrane	*E. coli* and *S. aureus* (120 μg/mL)	[121]

## Data Availability

Not applicable.

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
