# Peer review of "An Overview of Stimuli-Responsive Intelligent Antibacterial Nanomaterials"

_pharmaceutics, 2023, doi:10.3390/pharmaceutics15082113_

Round 1
Reviewer 1 Report
The review provides a captivating overview of the significance of stimuli-responsive intelligent antibacterial nanomaterials in addressing pathogenic bacteria and infectious diseases associated with biofilms. This topic holds immense relevance, given the escalating global health threat posed by antimicrobial resistance. The review effectively communicates the crucial aspects of the study. However, before the review can be accepted for publication, it requires to improve two key areas. Therefore, a major revision is recommended.
The conclusion should be rewritten to encompass not only a summarization of the materials outlined in the review but also the prospective trends and outlook for the future. The vision of the authors on this question is essential for the development of this emerging field. By providing insights into potential future directions and challenges, the conclusion will become more impactful and influential for researchers and practitioners.
The second issue lies in the fact that the authors have omitted information on antibacterial stimuli-responsive grafted coatings, except for reference 92, despite this trend gaining significant popularity in recent years.
Finally, I suggest the citation of highly relevant papers where stimuli-responsive antibacterial grafted coatings were intensely studied:
https://doi.org/10.1039/C9RA10874B
https://doi.org/10.1016/j.msec.2019.109806
https://doi.org/10.1021/la304708b
https://doi.org/10.1016/j.colsurfa.2022.128525
Minor editing of English language required.
Author Response
Response to Reviewer 1 Comments
The review provides a captivating overview of the significance of stimuli-responsive intelligent antibacterial nanomaterials in addressing pathogenic bacteria and infectious diseases associated with biofilms. This topic holds immense relevance, given the escalating global health threat posed by antimicrobial resistance. The review effectively communicates the crucial aspects of the study. However, before the review can be accepted for publication, it requires to improve two key areas. Therefore, a major revision is recommended.
The conclusion should be rewritten to encompass not only a summarization of the materials outlined in the review but also the prospective trends and outlook for the future. The vision of the authors on this question is essential for the development of this emerging field. By providing insights into potential future directions and challenges, the conclusion will become more impactful and influential for researchers and practitioners.
Response: Thank you for your valuable feedback and insightful comments on our manuscript. We truly appreciate the time and effort you have invested in reviewing our work. We acknowledge your suggestion regarding the conclusion of the paper. Your guidance is invaluable to us, and we are committed to producing a comprehensive and impactful paper. Thank you for your continued support and guidance throughout this review process.
The conclusion and outlook of the original manuscript have been reworked as follows:
The current challenges in antimicrobial research encompass antimicrobial resistance, biocompatibility and safety concerns, and the persistence and stability of antimicrobial materials. Drug-resistant bacterial have diminished the effectiveness of traditional antibiotics, highlighting the need for alternative antimicrobial strategies.
Extensively investigated stimuli-responsive antibacterial nanomaterials capable of responding to both endogenous and exogenous stimuli demonstrate significant potential in combating bacterial resistance. Compared to conventional nanocarriers, stimuli-responsive nanocarriers provide substantial benefits in achieving on-demand drug release, enhanced drug accumulation, and prolonged retention at infected sites, thanks to their sophisticated nanoplatform design, which enhances interactions with bacterial cells. Infected sites and bacterial biofilms have a distinct microenvironment that differs from normal tissues, featuring factors like low pH, overexpression of specific enzymes, high concentrations of H2O2, and specific toxin secretion. This specific microenvironment can be exploited to design endogenous stimuli-responsive nanoplatforms, enabling the targeted release of loaded bactericidal agents in response to the infected microenvironment, thereby enhancing the bioavailability, particularly for antibiotics. pH-responsive nanoplatforms have received more attention than other endogenous stimuli-responsive platforms for antibacterial applications, although challenges persist due to individual variations in endogenous stimuli. The pH levels at infected sites vary across different body parts, and infections caused by different bacterial strains may result in diverse microenvironments, with variations in secreted toxins and enzymes. Moreover, the degree of infection significantly influences the microenvironment. These factors should be considered when designing endogenous stimuli-responsive nanoplatforms. Exogenous stimuli-responsive nanoplatforms are equally important and have captured significant attention from researchers, especially in the realm of antibacterial applications. Light-responsive nanocarriers offer immense potential for achieving precise spatiotemporal drug release and pulsed delivery, presenting promising prospects for practical applications. The design of light-responsive nanomaterials stands at the forefront of current research, and the integration of composite light-responsive materials greatly enhances the application capabilities of light-responsive nanoparticles. Combined approaches, such as photo-responsive /antibiotic or photo-responsive/thermo-responsive combination therapies, exhibit substantial potential in eradicating bacterial infections. In addition, magnetism and ultrasound can also serve as exogenous stimuli for responsive nanoplatforms. Their greater penetration depths compared to light enable these stimuli to act as powerful release triggers for combating internal bacterial infections. The considerable potential of utilizing magnetism and ultrasound as exogenous stimuli for nanoplatforms calls for further research and investigation.
Despite the promising prospects of intelligent stimuli-responsive nanomaterials for antimicrobial applications, several challenges and issues need to be addressed during their development and widespread implementation:
(1) Safety Assessment: Comprehensive safety evaluations, including biocompatibility, toxicity, and metabolic pathways, are crucial for smart nanomaterials as novel agents. Ensuring their safety for human and environmental use is paramount, particularly for clinical applications.
(2) Lack of Standardization: The lack of uniform evaluation criteria and testing methods for smart nanomaterials results in incomparable outcomes and difficulties in generalizing their applications across different research institutions and countries.
(3) Antimicrobial Resistance: Although smart nanomaterials enable precise targeting of infection sites, rapid bacterial adaptability may lead to reduced antimicrobial efficacy and the emergence of resistance. Measures to counter antimicrobial resistance need to be implemented.
(4) Controlled Drug Release: Ensuring controlled drug release from smart nanomaterials is crucial to prevent either excessive or insufficient drug delivery, which could compromise treatment effectiveness.
(5) Large-Scale Production Costs: Complex manufacturing processes of smart nanomaterials might result in higher production costs. Further research and process optimization are required to lower costs and facilitate their mass production for clinical and industrial applications.
(6) Long-Term Stability: Ensuring the long-term stability of smart nanomaterials is vital, ensuring they retain their activity and do not undergo unpredictable changes during storage and use.
The future clinical prospects of smart responsive antimicrobial nanomaterials are highly promising. Intelligent responsiveness allows for personalized treatments tailored to individual needs and specific diseases, enhancing therapeutic efficacy while minimizing side effects. The application of smart responsive materials in medical devices, hospital surfaces, and wound management can effectively control infections and promote wound healing. This review presents a portion of examples, and there are many other fascinating works that have not been included. The field of stimulus-responsive intelligent nanomaterials is constantly evolving and optimizing. The examples provided here are just the tip of the iceberg, aimed at sparking interest and encouraging further exploration in this exciting area of research. We believe that smart responsive nano-antibacterial materials have a good application prospect in the future, and their biosafety, cost, yield and antibacterial effect will be recognized. In the future, the integration of nanomedicine with cutting-edge technologies such as artificial intelligence can be explored to construct more precise and intelligent disease diagnosis and treatment systems, opening up new directions for the development of biomedical science.
The second issue lies in the fact that the authors have omitted information on antibacterial stimuli-responsive grafted coatings, except for reference 92, despite this trend gaining significant popularity in recent years.
Finally, I suggest the citation of highly relevant papers where stimuli-responsive antibacterial grafted coatings were intensely studied:
https://doi.org/10.1039/C9RA10874B
https://doi.org/10.1016/j.msec.2019.109806
https://doi.org/10.1021/la304708b
https://doi.org/10.1016/j.colsurfa.2022.128525
Response 2: Thank you for your valuable feedback and for pointing out the issues in our manuscript. We sincerely appreciate your thorough review and constructive suggestions. Thank you for your continued support and dedication in reviewing our manuscript.
The section has been added as follows:
A series of temperature-responsive antibacterial polymer grafted doped silver nanoparticles nanomaterials have been successfully prepared [99-104]. These nanocomposites have excellent biocompatibility and exhibit changes in morphology and size in response to temperature variations. They can release drugs on demand. The incorporation of silver nanoparticles imparts outstanding antibacterial properties to the nanomaterial, inhibiting the proliferation of various bacteria. As a result, they hold great potential for development.

Reviewer 2 Report
This review aims to make an overview of the development of materials at the nanoscale presenting smart behavior and antibacterial properties. The manuscript seems interesting and has potential, but I believe needs several important improvements prior to being published in Pharmaceutics. Please, consider the following aspects:
1- Tables are very important to help the readership to follow and summarize aspects throughout the manuscript. I suggest the incorporation of some tables.
2- Figures. It needs to be introduced prior they appear in the manuscript. Also, I believe that this manuscript could be improved by incorporating figures from the authors and summarizing the most important findings of the literature.
3- Introduction. What do the authors refer to a global problem from the point of view of only one country's requirements? Lines 48-50.
4- Due to the manuscript being published in a journal devoted to pharmaceutics and the topic focusing on the application of nanomaterials as antibacterial agents, I believe the inclusion of the most relevant in vitro/in vivo results is needed. This fact is one of the weaknesses of the article. Also, light-mediated therapeutic strategies should be deeply explained.
5- Some interesting references should be considered for their addition. They could include recently published articles with similar topics in the field and other nanomaterials that have not been considered herein. I am listing below some of them: Materials Science and Engineering: C 107 (2020) 110334; Molecules 24,14 (2019) 2661; Biomedical Physics & Engineering Express 4,4 (2018) 045037; ACS Applied Nano Materials 5,8 (2022) 12019-12034; Materials Chemistry Frontiers (2023), 7, 355-380, Pharmaceutics (2023), 15(7), 1964; Pharmaceutics (2023) 15 (3), 809... ...
6- An abbreviation list/table is required in this kind of article for helping the readers.
7- The font size in Figure 3 is too little.
8- More discussions and summaries of the information with critical opinions are desirable throughout the whole manuscript.
Best regards,
Author Response
Response to Reviewer 2 Comments
This review aims to make an overview of the development of materials at the nanoscale presenting smart behavior and antibacterial properties. The manuscript seems interesting and has potential, but I believe needs several important improvements prior to being published in Pharmaceutics. Please, consider the following aspects:
Point 1:
Tables are very important to help the readership to follow and summarize aspects throughout the manuscript. I suggest the incorporation of some tables.
Response 1: Thank you for your valuable feedback and for highlighting the importance of tables in enhancing the readability and summarization of our manuscript. We truly appreciate your thoughtful review and constructive suggestions. Based on your recommendation, we have diligently worked on creating relevant tables that complement the text and present the data in a more organized and accessible format. The section has been added as follows:
Table 1. Examples of endogenous -stimulus responsive antibacterial nanomaterials
|
Antibacterial nanomaterials |
Triggers |
Nanocarriers |
Bactericidal moieties |
Bacteria/Biofilm |
ref |
|
AZM-DA NPs |
pH |
multi-segment graft copolymer |
azithromycin (AZM) |
in vitro: P. aeruginosa (AZM equivalent: 8 μg/mL) |
[43] |
|
Ag nanoparticle clusters (AgNCs) |
pH |
functional polymers |
release Ag+ |
MIC: MRSA (4 µg/ mL) and E. coli (8 µg/ mL) |
[46] |
|
PPEGMA-AuNRs |
pH |
Polymethacrylate (PCB) |
pH-induced surface charge-transformable, biofilm elimination |
MIC: E. coli and S. aureus (31.25 μg Au/mL) |
[47] |
|
ferulic acid-encapsulated nanoparticles (FA-NPs). |
pH |
amphiphile peptide |
ferulic acid |
MIC: E. coli (750 μg/mL), S. aureus (900 μg/mL) |
[56] |
|
AA@GS@HA-MNPs |
enzyme (hyaluronidase) |
graphene-mesoporous silica nanosheet@hyaluronic acid-magnetic nanoparticles |
ascorbic acid (AA) |
in vitro: E. coli and S. aureus (AA equivalent: 4 mg/mL) |
[44] |
|
Enzyme-responsive polymeric vesicles |
enzyme: penicillin G amidase (PGA) and β-lactamase (Bla) |
polymeric vesicles |
structural rearrangement and morphological transitions, release PGA and Bla |
MRSA, B. longum, L. acidophilus, and E. faecalis (1.0 μg/mL) |
[65] |
|
Ag-mesoporous silica nanoparticles (Ag-MONs) |
GSH |
mesoporous organosilica nanoparticles (MONs) |
Ag NPs |
E. coli and S. aureus. (1.28 μg/mL) |
[45] |
Table 2: Examples of exogenous -stimulus responsive antibacterial nanomaterials
|
Antibacterial nanomaterials |
Triggers |
Nanocarriers |
Bactericidal moieties |
Bacteria/Biofilm |
ref |
|
nanogel containing silver nanoparticles (AgNPs) |
light |
polycaprolactone (PCL) nanofibers mats |
release Ag+: disrupts ATP production and DNA replication |
S. aureus and E. coli (57.6 μg/mL) |
[89] |
|
DSPE-AuNR |
light |
Polymeric Hydrogel |
Photothermal-induced antibacterial activity |
P. aeruginosa biofilm (0.25–0.03 nM) |
[91] |
|
ZnTCPP@ZnO |
ultrasound |
MOF |
ROS |
Propionibacterium acnes |
[107] |
|
BM2-LVFX-NPs |
ultrasound |
levofloxacin-loaded PLGA-PEG nanoparticles with BM2 aptamer |
ROS |
Bacillus Calmette-Guérin bacteria |
[110] |
|
RBC-HNTM-Pt@Au |
ultrasound |
Au NRs-actuated single-atom-doped porphyrin MOF (HNTM-Pt@Au) |
ROS, dynamically neutralize the secreted toxins |
MRSA |
[111] |
|
HNTM/Nb2C |
ultrasound |
Nb2C nanosheet-decorated porphyrin MOF hollow nanotubes (HNTM/Nb2C) |
ROS (the rapid charge transfer and suppressed recombination of electron-hole pairs) |
MRSA |
[113] |
|
a Na+ inserted PB system |
Microwave |
MOF, Prussian blue (PB) |
Fenton reaction and thermal effects |
E. coli and S. aureus |
[114] |
|
TiO2@MS-SH/Nor-Dex nanocomposite hydrogels |
ultrasound |
mesoporous silica-coated TiO2 nanoparticles with thiolated surface functionalization (TiO2@MS-SH), norbornene-functionalized dextran (Nor-Dex) |
ROS |
E. coli and S. aureus |
[115] |
|
NCJC/Fe3O4/Ag |
magnetic field |
Nanocrystalline jute cellulose (NCJC) particles |
Ag+ |
S. aureus, E. coli, S. dysenteriae, S. boydii, Shigella boydii (5 μg/mL) |
[117] |
|
TA-CFO/PVA |
magnetic field |
tannin (TA), cobalt ferrite nanoparticles (CFO NPs), polyvinyl alcohol (PVA) matrix |
TA, Co2+ |
E. coli and S. aureus |
[118] |
|
GLM-Fe |
magnetic field |
Galinstan-based liquid-metal microparticles and nanoparticles (GLM-Fe) |
magnetic field induces the GLM-Fe particles to spin, shape-transform, and impart physical forces to the bacteria |
P. aeruginosa and S. aureus (100 μg/mL) |
[120] |
|
MXene@Fe3O4/Au/PDA nanosheets |
magnetic field/light |
polydopamine |
PTT, nanosheet cuts the cytomembrane |
E. coli and S. aureus (120 μg/mL) |
[121] |
This review only presents a portion of examples, and there are many other fascinating works that have not been included. The field of stimulus-responsive intelligent nanomaterials is constantly evolving and optimizing. The examples provided here are just the tip of the iceberg, aimed at sparking interest and encouraging further exploration in this exciting area of research.
Point 2:
Figures. It needs to be introduced prior they appear in the manuscript. Also, I believe that this manuscript could be improved by incorporating figures from the authors and summarizing the most important findings of the literature.
Response 2: Thank you very much for reviewing our paper and providing valuable feedback. We acknowledge your suggestions regarding this particular section, and we have made corresponding adjustments and improvements throughout the entire manuscript. Instead of directly copying and pasting the content here, we opted to make unified changes throughout the paper to better reflect the modifications. Throughout the entire paper, we carefully examined and revised the content based on your suggestions to enhance its logical flow, coherence, and accuracy. We believe that this approach allows us to integrate the modifications more effectively and make the paper more polished. Once again, we sincerely appreciate your patient review and invaluable recommendations. We genuinely hope that these revisions will enhance the quality of the paper. If there are any other areas that require improvement, please let us know, and we will promptly make the necessary adjustments.
Point 3:
Introduction. What do the authors refer to a global problem from the point of view of only one country's requirements? Lines 48-50.
Response 3: Thank you for your valuable feedback and for bringing up the concern regarding the global problem mentioned in our introduction. We genuinely appreciate your thorough review and insightful comments. We understand your point about the need to consider the issue from a broader perspective rather than solely focusing on one country's requirements. Upon reevaluating our introduction (Lines 48-50), we agree that it is essential to acknowledge the global implications of the problem we are addressing. To address this concern, we have revised the introduction to provide a more comprehensive outlook that recognizes the global nature of the issue and its impact on various regions and communities worldwide. By doing so, we aim to present a more inclusive and accurate representation of the problem's significance on a global scale. We genuinely value your input and recognize the importance of approaching our research with a broader perspective. Your feedback will undoubtedly contribute to the overall quality and relevance of our manuscript.
The section has been revised as follows:
Superbug that are resistant to antibiotics will seriously reduce the effectiveness of people's antibacterial methods based on antibiotics and lead to a public health crisis. Many countries are consciously raising public awareness of antibiotics, strictly regulating the scope and process of antibiotic use, and preventing the misuse or improper use of antibiotics.
Point 4:
Due to the manuscript being published in a journal devoted to pharmaceutics and the topic focusing on the application of nanomaterials as antibacterial agents, I believe the inclusion of the most relevant in vitro/in vivo results is needed. This fact is one of the weaknesses of the article. Also, light-mediated therapeutic strategies should be deeply explained.
Response 4: Thank you for your valuable feedback and for bringing up the important points regarding the content of our manuscript. We genuinely appreciate your thorough review and constructive comments. In our study, we focused on the dynamic changes of responsive nanomaterials as antibacterial drugs under endogenous or exogenous stimuli, ignoring the safety, metabolism and kinetic behavior of the drugs in vitro or in vivo. In our conclusion section, we mention that one of the challenges of responsive nanomaterials is to ensure the long-term biosafety and stability of antibacterial nanomaterials. The section has been revised as follows:
“Despite the promising prospects of intelligent stimuli-responsive nanomaterials for antimicrobial applications, several challenges and issues need to be addressed during their development and widespread implementation:
(1) Safety Assessment: Comprehensive safety evaluations, including biocompatibility, toxicity, and metabolic pathways, are crucial for smart nanomaterials as novel agents. Ensuring their safety for human and environmental use is paramount, particularly for clinical applications.
(2) Lack of Standardization: The lack of uniform evaluation criteria and testing methods for smart nanomaterials results in incomparable outcomes and difficulties in generalizing their applications across different research institutions and countries.
(3) Antimicrobial Resistance: Although smart nanomaterials enable precise targeting of infection sites, rapid bacterial adaptability may lead to reduced antimicrobial efficacy and the emergence of resistance. Measures to counter antimicrobial resistance need to be implemented.
(4) Controlled Drug Release: Ensuring controlled drug release from smart nanomaterials is crucial to prevent either excessive or insufficient drug delivery, which could compromise treatment effectiveness.
(5) Large-Scale Production Costs: Complex manufacturing processes of smart nanomaterials might result in higher production costs. Further research and process optimization are required to lower costs and facilitate their mass production for clinical and industrial applications.
(6) Long-Term Stability: Ensuring the long-term stability of smart nanomaterials is vital, ensuring they retain their activity and do not undergo unpredictable changes during storage and use.”
In addition, we deeply explained the light-mediated therapeutic strategies, The section has been revised as follows:
“Photo-responsive antibacterial nanomaterials usually consist of chromophores that can transform light inputs into potential chemical or energy outputs, whether based on their intrinsic optical properties or in conjunction with photosensitizers and photothermal agents, play a critical role in light-mediated therapeutic strategies. These strategies encompass a range of techniques, including photothermal therapy (PTT), photodynamic therapy (PDT). PTT is a thermal-based therapy technique that utilizes nanomaterials with high photothermal conversion efficiency to convert light energy into heat energy when exposed to an external light source. High temperatures generated by photothermal effects (>42°C) can induce bacterial cell apoptosis, disrupt cell membranes, damage the cell cytoskeleton, and inhibit DNA synthesis. PDT has emerged as a promising approach to effectively eradicate pathogenic bacteria by utilizing a photosensitizer and laser irradiation. PDT oxidizes biomolecules and inflicts irreversible damage through the generation of reactive oxygen species (ROS). A notable advantage of PDT over conventional antibiotic-based therapies is its capacity for repeated treatment without inducing undesirable drug resistance. PTT and PDT has become a promising antibacterial method due to its low invasiveness, low toxicity and avoidance of drug-resistant bacteria.”
Point 5:
Some interesting references should be considered for their addition. They could include recently published articles with similar topics in the field and other nanomaterials that have not been considered herein. I am listing below some of them: Materials Science and Engineering: C 107 (2020) 110334; Molecules 24,14 (2019) 2661; Biomedical Physics & Engineering Express 4,4 (2018) 045037; ACS Applied Nano Materials 5,8 (2022) 12019-12034; Materials Chemistry Frontiers (2023), 7, 355-380, Pharmaceutics (2023), 15(7), 1964; Pharmaceutics (2023) 15 (3), 809... ...
Response 5: Thank you for your feedback. We have now added the relevant references to the appropriate sections in the revised manuscript. We appreciate your careful evaluation and are committed to ensuring that all necessary citations are properly included. Instead of directly copying and pasting the content here, we opted to make unified changes throughout the paper to better reflect the modifications.
Point 6:
An abbreviation list/table is required in this kind of article for helping the readers.
Response 6: Thank you for your suggestion. We have now included the abbreviation list in the appropriate section of the revised manuscript. We appreciate your thoughtful feedback and are committed to improving the manuscript's clarity and readability.
|
Abbreviations |
|
|
ROS |
reactive oxygen species |
|
EPS |
extracellular polymeric substances |
|
NPs |
nanoparticles |
|
Hydase |
hyaluronidase |
|
PGA |
penicillin G amidase |
|
βla |
β-lactamase |
|
CBS |
cystathionine β-synthase |
|
CSE |
cystathionine γ-lyase |
|
PMB |
Polymyxin B |
|
PDA |
polydopamine |
|
GSH |
glutathione |
|
HA |
hyaluronic acid |
|
PDT |
photodynamic therapy |
|
PTT |
photothermal therapy |
|
US |
ultrasound |
|
MW |
microwaves |
|
H2O2 |
hydrogen peroxide |
|
O2- |
superoxide anions |
|
·OH |
hydroxyl radicals |
|
P. aeruginosa |
Pseudomonas aeruginosa |
|
MRSA |
Methicillin-resistant Staphylococcus aureus |
|
E. coli |
Escherichia coli |
|
S. aureus |
Staphylococcus aureus |
|
B. longum |
Bifidobacterium longum |
|
L. acidophilus |
Lactobacillus acidophilus |
|
E. faecalis |
Enterococcus faecalis |
Point 7:
The font size in Figure 3 is too little.
Response 7:
Thank you for your sincere suggestion. Figure 3 is an image obtained from a referenced paper. Considering that the font size might be too little, I have made certain adjustments to Figure 3 to ensure that the text is clear and visible.
Point 8:
More discussions and summaries of the information with critical opinions are desirable throughout the whole manuscript.
Response 8:
Thank you for your valuable feedback. We appreciate your suggestion for including more discussions and critical opinions throughout the manuscript. In the revised version, we will carefully address this point by enhancing the discussions and providing more comprehensive summaries of the information presented. We will strive to incorporate critical opinions and insights to offer a deeper analysis of our findings and their implications.
In the second section of Formation of bacterial resistance and antibacterial strategies, we added more discussions and summaries as follows:
“As an essential component in all bacterial cells, the bacterial outer membrane or cell wall is responsible for the maintenance of cell shape, osmotic regulation, protection against mechanical stress, and combating infection. Physical/mechanical damage of the bacterial outer membrane or cell wall (Figure 2a) can lead to its dysfunction and leakage of cytoplasmic components to finally cause bacteriostatic and bactericidal effects. This pathway has been regarded as one of the most common antibacterial mechanism.”
In the third section of Characteristics and mechanisms of antibacterial nanomaterials, we added more discussions and summaries as follows:
“Nanomaterial-based therapies are promising tools to combat bacterial infections that are difficult to treat, featuring the capacity to evade existing mechanisms associated with acquired drug resistance. In addition, the unique size and physical properties of nanomaterials give them the capability to target biofilms, overcoming recalcitrant infections. The design and synthesis of antibacterial nanomaterials (NM) with improved efficiency, stability, fouling resistance, biocompatibility and recyclability is a fast-developing field.”
In the fourth section, we added more discussions and summaries as follows:
“Nanomaterials that respond to bacterial metabolites stimuli are favoring self-adaptive antibacterial systems and precision medicine. Bactericides can be precisely released or exposed on demand, that is, only when and where needed, which can avoid the over-use of bactericides and reduce the generation of drug-resistant bacteria.”
In the section of 4.1, we added more discussions and summaries as follows:
“It is known that the infected tissues and bacterial biofilms have specific microenvironment, which is different from the normal tissues, such as low pH, upregulated enzymes, high reactive oxygen species, and so forth. The specific microenvironment can be used to endogenously trigger specific properties of the nanoparticles, such as drug release, charge reversal, size change, and so on. In theory, the stimuli responsive behavior of the nanoparticles can only be triggered upon arriving at the infected site, which is very advantageous in improving the drug bioavailability.”
In the section of 4.1.1, we added more discussions and summaries as follows:
“Based on nanomaterials or their surface properties, the pH-triggered response could be manifested as pH-triggered drug release and surface charge reversal. By leveraging the acidic environment specific to the site of bacterial infection, researchers can design targeted and controlled release systems or specific antibacterial effects in the treatment of bacterial infections. The mechanism of the pH-responsiveness can be divided into two categories: (1) protonation/deprotonation of amine groups and carboxyl groups; (2) cleavage of chemical bond. The protonation of amine groups is a frequently used strategy to fabricate pH-responsive nanoplatforms. Therefore, various pH-responsive antibacterial nanomaterials were reported to improve the therapeutic performance in treating bacterial infections.”
In the section of 4.1.2, we added more discussions and summaries as follows:
“In addition to the acidic microenvironment in bacterial infected sites, the bacteria can secret various enzymes during proliferation, such as hyaluronidase, lipase, phosphatase, phospholipase, gelatinase, matrix metalloproteinase, and protease, which constitute a unique microenvironment in the infected sites”
In the section of 4.2, we added more discussions and summaries as follows:
“The exogenous stimuli-responsive nanoparticles can be triggered by external stimuli, such as light, magnetic field, electric field, ultrasound, and so forth. Since the external stimuli can be easily controlled, the exogenous stimuli-responsive nanoplatforms are very promising to achieve spatiotemporally controlled drug delivery. Besides, responsive antibacterial surface also needs delicate design of stimuli-responsive materials.
Although various endogenous stimuli-responsive nanoplatforms were fabricated, it is still very difficult to realize accurately controlled drug release due to the complicated physiological environment and large individual differences. It would be favorable to develop exogeneous stimuli-responsive nanoplatforms since the exogeneous stimuli can be easily controlled.”
In the section of 4.2.2, we made more discussions and summaries as follows:
“Thermally responsive nanomaterials are a distinct type of material different from light-thermal nanomaterials, which are considered another type of "smart" nanomaterials because they are able to change their physical properties (e.g. shape, size, etc.) in response to mild temperature changes, triggering the controlled release of drug . Thermo-responsive polymers exhibit a phase change at their critical solution temperature and such behavior can be attributed to disruption of intra and intermolecular interactions that cause the polymer to either expand or collapse within the aqueous solvent. Polymers with a lower critical solution temperature (LCST) will display phase separation (e.g., precipitation) above a specific temperature, while those with an upper critical solution temperature will display phase separation (e.g., precipitation) below a specific temperature. In hydrogel systems, where there is an increased or decreased swelling around this transition temperature, investigators sometimes refer to this swelling transition as the volume phase transition temperature.
Thermo-responsive materials can be divided into two categories, water-insoluble above the lower critical solution temperature (LCST) and water-insoluble below the upper critical solution temperature. The materials that have transition temperature close to body temperature are widely used as thermo-responsive materials in biomedical field. Among different thermo-responsive materials, poly(n-isopropylacrylamide) (pNIPAM) is the most commonly used thermo-responsive polymer, which can transform reversibly from a hydrophilic coil state to a hydrophobic globule state near 32 °C by varying the polymer concentration and incorporating surfactants and copolymers. pNIPAM is also frequently crosslinked to create hydrogels with reversible thermal shrinkage and expansion properties. pNIPAM is easily functionalized and has been used as a hydrophilic segment in the synthesis of thermoresponsive amphiphilic block copolymers. Therefore, pNIPAM is an excellent polymer for developing materials with controllable antimicrobial properties.”
In the section of Conclusion, we made more discussions and summaries as follows:
“The current challenges in antimicrobial research encompass antimicrobial resistance, biocompatibility and safety concerns, and the persistence and stability of antimicrobial materials. Drug-resistant bacterial have diminished the effectiveness of traditional antibiotics, highlighting the need for alternative antimicrobial strategies.
Extensively investigated stimuli-responsive antibacterial nanomaterials capable of responding to both endogenous and exogenous stimuli demonstrate significant potential in combating bacterial resistance. Compared to conventional nanocarriers, stimuli-responsive nanocarriers provide substantial benefits in achieving on-demand drug release, enhanced drug accumulation, and prolonged retention at infected sites, thanks to their sophisticated nanoplatform design, which enhances interactions with bacterial cells. Infected sites and bacterial biofilms have a distinct microenvironment that differs from normal tissues, featuring factors like low pH, overexpression of specific enzymes, high concentrations of H2O2, and specific toxin secretion. This specific microenvironment can be exploited to design endogenous stimuli-responsive nanoplatforms, enabling the targeted release of loaded bactericidal agents in response to the infected microenvironment, thereby enhancing the bioavailability, particularly for antibiotics. pH-responsive nanoplatforms have received more attention than other endogenous stimuli-responsive platforms for antibacterial applications, although challenges persist due to individual variations in endogenous stimuli. The pH levels at infected sites vary across different body parts, and infections caused by different bacterial strains may result in diverse microenvironments, with variations in secreted toxins and enzymes. Moreover, the degree of infection significantly influences the microenvironment. These factors should be considered when designing endogenous stimuli-responsive nanoplatforms. Exogenous stimuli-responsive nanoplatforms are equally important and have captured significant attention from researchers, especially in the realm of antibacterial applications. Light-responsive nanocarriers offer immense potential for achieving precise spatiotemporal drug release and pulsed delivery, presenting promising prospects for practical applications. The design of light-responsive nanomaterials stands at the forefront of current research, and the integration of composite light-responsive materials greatly enhances the application capabilities of light-responsive nanoparticles. Combined approaches, such as photo-responsive /antibiotic or photo-responsive/thermo-responsive combination therapies, exhibit substantial potential in eradicating bacterial infections. In addition, magnetism and ultrasound can also serve as exogenous stimuli for responsive nanoplatforms. Their greater penetration depths compared to light enable these stimuli to act as powerful release triggers for combating internal bacterial infections. The considerable potential of utilizing magnetism and ultrasound as exogenous stimuli for nanoplatforms calls for further research and investigation.
Despite the promising prospects of intelligent stimuli-responsive nanomaterials for antimicrobial applications, several challenges and issues need to be addressed during their development and widespread implementation:
(1) Safety Assessment: Comprehensive safety evaluations, including biocompatibility, toxicity, and metabolic pathways, are crucial for smart nanomaterials as novel agents. Ensuring their safety for human and environmental use is paramount, particularly for clinical applications.
(2) Lack of Standardization: The lack of uniform evaluation criteria and testing methods for smart nanomaterials results in incomparable outcomes and difficulties in generalizing their applications across different research institutions and countries.
(3) Antimicrobial Resistance: Although smart nanomaterials enable precise targeting of infection sites, rapid bacterial adaptability may lead to reduced antimicrobial efficacy and the emergence of resistance. Measures to counter antimicrobial resistance need to be implemented.
(4) Controlled Drug Release: Ensuring controlled drug release from smart nanomaterials is crucial to prevent either excessive or insufficient drug delivery, which could compromise treatment effectiveness.
(5) Large-Scale Production Costs: Complex manufacturing processes of smart nanomaterials might result in higher production costs. Further research and process optimization are required to lower costs and facilitate their mass production for clinical and industrial applications.
(6) Long-Term Stability: Ensuring the long-term stability of smart nanomaterials is vital, ensuring they retain their activity and do not undergo unpredictable changes during storage and use.
The future clinical prospects of smart responsive antimicrobial nanomaterials are highly promising. Intelligent responsiveness allows for personalized treatments tailored to individual needs and specific diseases, enhancing therapeutic efficacy while minimizing side effects. The application of smart responsive materials in medical devices, hospital surfaces, and wound management can effectively control infections and promote wound healing. We believe that smart responsive nano-antibacterial materials have a good application prospect in the future, and their biosafety, cost, yield and antibacterial effect will be recognized. In the future, the integration of nanomedicine with cutting-edge technologies such as artificial intelligence can be explored to construct more precise and intelligent disease diagnosis and treatment systems, opening up new directions for the development of biomedical science.”

Reviewer 3 Report
It is very difficult to judge review articles. Given that such articles are based on already published scientific articles that have passed a strict review process, it is not necessary to assess their scientific quality, and the more important aspects are logical structure, clarity and a broader perspective enabling the reader to gain an overview of the given topic.
From my point of view, the reviewed article does not fully meet these requirements. I see as a main problem that the authors focused mainly on the description of the methods leading to the production of selected types of antibacterial nanomaterials and not on the description and categorization/classification of their antibacterial action. In principle, the authors do not carefully define the term nanomaterial and its possible modes of action in the text. In my opinion, it should be clearly distinguished if the nanomaterial itself acts as i) an active antibacterial agent, ii) a carrier of an antibacterial substance or iii) change the environment upon external stimuli, e.g., heating during irradiation. The actual antibacterial mechanisms, which strongly depend on the properties of nanomaterials (inorganic, polymer, nanoparticles, nanorods, etc.) are thus more or less only indicated in Figure 3 without any deeper discussion and description. This subsequently also applies to the modes of action of individual stimuli-responsive nanomaterials. As an example, it is possible to use the very first of them, i.e., pH-responsive materials (section 4.1.1). Here, the authors describe in detail 4 different types of nanomaterials and their synthesis. In my opinion, it would be more interesting for the reader to describe different possible strategies, i.e., in this particular case, to divide the possible strategies into different groups that are based on i) change in the surface charge of nanomaterials in an acidic environment, which enables the effective binding of nanomaterials to bacteria, ii) cleavage of bonds in the acidic environment followed by the release of antibacterial agent and iii) conformation/morphology change with pH variation that facilitates penetration of nanomaterials into the bacteria. This I found more important than giving the details of the selected fabrication process that interested readers may find in cited works. Such structurization of the text, which is used solely in the section devoted to magnetic-responsive materials, could increase the readability and importance of the article. In addition, the discussion of the antibacterial effect (its efficiency towards a studied pathogen, duration of the antibacterial effect, selectivity, etc.) of described nanomaterials is often missing. For instance, it is only stated that “Moreover, the material exhibits high antibacterial efficacy (figure 5a).” (line 302), but it is not clear what this means and figure 5 does not show biological results but only the structure of the used nanoparticle. To conclude, instead of focusing on the different principles behind the action of the stimuli-responsive antibacterial nanomaterials and discussing their antibacterial efficiency, the authors present a collection of examples of different synthesis methods of stimuli-responsive antibacterial nanomaterials. Despite these criticisms, it is important to emphasize that the above is only my personal opinion, not intended to impugn the authors' work in any way, and should be seen as a well-intentioned suggestion for possible changes to the manuscript to improve its quality.
Besides the above given general comment, I have several, less crucial ones:
Some sentences are hard to read, incomplete or repetitive and the text should be carefully reformulated. Selected examples:
• line 325 “In the study conducted by Li et al [66]. As depicted…”
• compare lines 258 “Once passive targeting to the acidic environment of the infection site, the positively charged guanidine groups become exposed on the surface of CGNs” and 263 “Upon passive targeting to the acidic environment of the infection site, the positively charged guanidine groups in CGNs are exposed, ..”
• compare lines 261 “Subsequently, the encapsulated CS (chitosan) in CGNs generates a considerable amount of heat under near-infrared (NIR) irradiation, thereby effectively eliminating biofilms” and line 265 “Subsequently, under NIR irradiation, the encapsulated CS in CGNs generates significant heat, leading to the efficient eradication of biofilms (Figure 4c)”
It is not clear to me why H2S is termed as ROS – line 336 “by elevated levels of reactive oxygen species (ROS), such as H2O2 or H2S...”
The material presented in figure 6b is pH-responsive and not gas-responsive and thus is in the wrong section.
Some terms should be explained to the non-specialist, e.g., “upconversion”, “hot ions effect”...
Authors should unify citing style ( Author at al. [] X Author et al []) and used abbreviations, e.g., PNIPAM and pNIPAM
Author Response
Response to Reviewer 3 Comments
It is very difficult to judge review articles. Given that such articles are based on already published scientific articles that have passed a strict review process, it is not necessary to assess their scientific quality, and the more important aspects are logical structure, clarity and a broader perspective enabling the reader to gain an overview of the given topic.
From my point of view, the reviewed article does not fully meet these requirements. I see as a main problem that the authors focused mainly on the description of the methods leading to the production of selected types of antibacterial nanomaterials and not on the description and categorization/classification of their antibacterial action. In principle, the authors do not carefully define the term nanomaterial and its possible modes of action in the text. In my opinion, it should be clearly distinguished if the nanomaterial itself acts as i) an active antibacterial agent, ii) a carrier of an antibacterial substance or iii) change the environment upon external stimuli, e.g., heating during irradiation. The actual antibacterial mechanisms, which strongly depend on the properties of nanomaterials (inorganic, polymer, nanoparticles, nanorods, etc.) are thus more or less only indicated in Figure 3 without any deeper discussion and description. This subsequently also applies to the modes of action of individual stimuli-responsive nanomaterials. As an example, it is possible to use the very first of them, i.e., pH-responsive materials (section 4.1.1). Here, the authors describe in detail 4 different types of nanomaterials and their synthesis. In my opinion, it would be more interesting for the reader to describe different possible strategies, i.e., in this particular case, to divide the possible strategies into different groups that are based on i) change in the surface charge of nanomaterials in an acidic environment, which enables the effective binding of nanomaterials to bacteria, ii) cleavage of bonds in the acidic environment followed by the release of antibacterial agent and iii) conformation/morphology change with pH variation that facilitates penetration of nanomaterials into the bacteria. This I found more important than giving the details of the selected fabrication process that interested readers may find in cited works. Such structurization of the text, which is used solely in the section devoted to magnetic-responsive materials, could increase the readability and importance of the article. In addition, the discussion of the antibacterial effect (its efficiency towards a studied pathogen, duration of the antibacterial effect, selectivity, etc.) of described nanomaterials is often missing. For instance, it is only stated that “Moreover, the material exhibits high antibacterial efficacy (figure 5a).” (line 302), but it is not clear what this means and figure 5 does not show biological results but only the structure of the used nanoparticle. To conclude, instead of focusing on the different principles behind the action of the stimuli-responsive antibacterial nanomaterials and discussing their antibacterial efficiency, the authors present a collection of examples of different synthesis methods of stimuli-responsive antibacterial nanomaterials. Despite these criticisms, it is important to emphasize that the above is only my personal opinion, not intended to impugn the authors' work in any way, and should be seen as a well-intentioned suggestion for possible changes to the manuscript to improve its quality.
Response:
Thank you for your valuable feedback. In our manuscript, we focused on describing the types of antibacterial nanomaterials in response to stimuli, while ignoring the antibacterial mechanism and antibacterial efficiency of specific antibacterial nanomaterials. we only made a broad introduction to the antibacterial mechanism of antibacterial nanomaterials in the third section. In the fourth section, considering the emphasis on the dynamic changes made by antibacterial nanomaterials in response to stimuli, the antibacterial mechanism of antibacterial nanomaterials is rarely described, but it belongs to the scope of antibacterial mechanism in the third part. For a specific antibacterial material, its antibacterial mode may be an antimicrobial carrier, or it may be an active antibacterial agent itself, or it may have both, so there is no specific description of the antibacterial mode in this paper. We genuinely appreciate your thorough review and insightful comments., we have added more discussion and summary in the revised manuscript.
In the second section of Formation of bacterial resistance and antibacterial strategies, we added more discussions and summaries as follows:
“The bacterial outer membrane or cell wall, being a crucial component in all bacterial cells, plays a vital role in maintaining cell shape, osmotic regulation, protection against mechanical stress, and defense against infection. Physical or mechanical damage to the bacterial outer membrane or cell wall (Figure 2a) can result in its dysfunction and the subsequent leakage of cytoplasmic components, ultimately leading to bacteriostatic and bactericidal effects. This mechanism is widely acknowledged as one of the most common antibacterial pathways.”
In the third section of Characteristics and mechanisms of antibacterial nanomaterials, we added more discussions and summaries as follows:
“Therapies based on nanomaterials hold great promise in combating challenging bacterial infections, as they have the ability to circumvent acquired drug resistance mechanisms. Moreover, nanomaterials possess unique size and physical properties that enable them to target biofilms, effectively addressing recalcitrant infections. The design and synthesis of antibacterial nanomaterials (NM) with enhanced efficiency, stability, fouling resistance, biocompatibility, and recyclability represent a rapidly advancing field.”
In the fourth section, we added more discussions and summaries as follows:
“Nanomaterials responsive to bacterial metabolite stimuli are paving the way for the development of precision medicine and self-adaptive antibacterial systems. Bacteri-cides can be exposed or precisely released on demand. Indeed, this approach holds sig-nificant promise in addressing the problem of overusing bactericides and mitigating the emergence of drug-resistant bacteria.”
In the section of 4.1, we added more discussions and summaries as follows:
“The infected tissues and bacterial biofilms are known to possess a specific microenvironment that differs from normal tissues. This microenvironment is characterized by factors such as low pH, high reactive oxygen species, upregulated enzymes, and more. The unique microenvironment in infected tissues and bacterial biofilms can be exploited to internally trigger specific properties of nanoparticles, including drug release, charge reversal, size change, and other functionalities. The stimuli-responsive behavior of nanoparticles offers a theoretical advantage, as it can be specifically activated upon reaching the infected site, providing a significant advantage in enhancing drug bioavailability.”
In the section of 4.1.1, we added more discussions and summaries as follows:
“The mechanism of pH-responsiveness can be categorized into two main groups: (1) protonation/deprotonation of amine groups and carboxyl groups, and (2) cleavage of chemical bonds. Protonation of amine groups is a commonly employed approach to create pH-responsive nanoplatforms. As a result, numerous pH-responsive antibacterial nanomaterials have been reported, showing enhanced therapeutic efficacy in the treatment of bacterial infections.”
In the section of 4.1.2, we added more discussions and summaries as follows:
“Enzyme secretion is another characteristic of the site of bacterial infection is another characteristic of the site of bacterial infection. These enzymes create a unique microenvironment at the site of infection. Biomaterials such as HA, poly (ε-caprolactone), and polyphosphates can be degraded by these enzymes, leading to the release of encapsulated antibacterial agents. As a result, nanotargeted antibacterial therapy with such encapsulated antibacterial agents has garnered increasing attention.”
In the section of 4.2, we added more discussions and summaries as follows:
“Exogenous stimuli-responsive nanoparticles can be activated by external stimuli, such as light, magnetic fields, electric fields, ultrasound, and more. Given the ease of controlling these external stimuli, exogenous stimuli-responsive nanoplatforms hold great promise in achieving spatiotemporally controlled drug delivery. Additionally, the design of responsive antibacterial surfaces requires careful consideration of stimuli-responsive materials.
Despite the fabrication of various endogenous stimuli-responsive nanoplatforms, achieving accurate control over drug release remains challenging due to the complex physiological environment and significant individual differences. In this context, the development of exogenous stimuli-responsive nanoplatforms becomes advantageous, as external stimuli can be readily controlled.”
In the section of 4.2.2, we made more discussions and summaries as follows:
“Thermo-responsive polymers experience a phase transition at their critical solution temperature, resulting from the disturbance of intra and intermolecular interactions. This behavior leads the polymer to either expand or collapse within the aqueous solvent. Polymers with a lower critical solution temperature (LCST) demonstrate phase separation, such as precipitation, when the temperature exceeds a specific threshold. Conversely, polymers with an upper critical solution temperature display phase separation below a specific temperature. In hydrogel systems, where the swelling increases or decreases around this transition temperature, researchers sometimes refer to this swelling transition as the volume phase transition temperature.”
Besides the above given general comment, I have several, less crucial ones:
Some sentences are hard to read, incomplete or repetitive and the text should be carefully reformulated. Selected examples:
line 325 “In the study conducted by Li et al [66]. As depicted…”
Response: Thank you for your suggestion. We have now rephrased the passage using more appropriate language. We appreciate your valuable feedback and are committed to continuously improving the clarity and quality of our manuscript. This sentence has been modified in the revised manuscript as follows:.
“In another study, Li et al. successfully fabricated penicillin G amidase (PGA) and β-lactamase (Bla)-responsive polymeric vesicles, as shown in Figure 5b…”
compare lines 258 “Once passive targeting to the acidic environment of the infection site, the positively charged guanidine groups become exposed on the surface of CGNs” and 263 “Upon passive targeting to the acidic environment of the infection site, the positively charged guanidine groups in CGNs are exposed, …” compare lines 261 “Subsequently, the encapsulated CS (chitosan) in CGNs generates a considerable amount of heat under near-infrared (NIR) irradiation, thereby effectively eliminating biofilms” and line 265 “Subsequently, under NIR irradiation, the encapsulated CS in CGNs generates significant heat, leading to the efficient eradication of biofilms (Figure 4c)”
Response: We apologize for the oversight and the repetitive statement in the manuscript during the proofreading process. In the revised version, we have taken the necessary corrections and removed this error from the manuscript. Thank you for bringing this to our attention, and we appreciate your understanding. These sentences have been modified in the revised manuscript as follows:.
“Upon passive targeting to the acidic environment of the infection site, the positively charged guanidine groups in CGNs are exposed, imparting CGNs with deep penetration and efficient bacterial adhesion. Subsequently, under NIR irradiation, the encapsulated CS in CGNs generates significant heat, leading to the efficient eradication of biofilms”
It is not clear to me why H2S is termed as ROS – line 336 “by elevated levels of reactive oxygen species (ROS), such as H2O2 or H2S...”
Response: We apologize for any inconvenience caused by our inaccurate description, which may have hindered your reading. We have provided a more precise and accurate representation in the revised manuscript. This sentence has been modified in the revised manuscript as follows:.
“Bacterial infection sites are frequently accompanied by elevated levels of H2O2, H2S, which play a crucial role in the pathogenesis of infections"
The material presented in figure 6b is pH-responsive and not gas-responsive and thus is in the wrong section.
Response: Thank you for your professional feedback. In the revised manuscript, we have made necessary adjustments and recited more suitable references. We understand the importance of accurate classification and have rechecked the manuscript to ensure that all figure is placed in their respective relevant chapters. The section has been revised as follows:
“Guo et al. proposed a spatially selective chemodynamic therapy strategy using bimetallic nanomaterials responsive to the microenvironment (pH and H2O2). Within the acidic microenvironment of the biofilm, CuFe5O8 nanocubes act as efficient catalysts, promoting the generation of abundant ·OH radicals, which effectively cleave extracellular DNA and disrupt the dense biofilm structure (Figure 6b). Moreover, in the extracellular environment outside the biofilm (characterized by higher pH and lower H2O2 concentration), CuFe5O8 nanocubes generate a controlled amount of ·OH radicals, leading to the reversal of the immunosuppressive microenvironment surrounding the biofilm. This process effectively activates local macrophages and facilitates the clearance of damaged biofilms and floating bacteria”
Some terms should be explained to the non-specialist, e.g., “upconversion”, “hot ions effect”.
Response: Thank you for your valuable feedback. We appreciate your suggestion to explain certain terms to make the manuscript easier for non-professional readers to read. In the revised manuscript, we provide detailed explanations and definitions of terms such as "upconversion" and "thermionic effect" to ensure that all readers, including those who may not be familiar with a specific field, can clearly understand and understand them. Your suggestion will undoubtedly improve the overall readability and inclusiveness of the manuscript. Thank you again for taking the time and thoughtful evaluation of our work.
“Upconversion” has been revised as follows:
“upconversion nanoparticles (UCNP, which are nanoscale particles capable of converting lower-energy light into higher-energy light through nonlinear optical processes.)…”
“hot ions effect” has been revised as follows:
“By leveraging the unique "hot ions effect (some ions in the plasma have higher energy than ordinary ions, and these high-energy ions damage the cell wall and membrane of microorganisms, resulting in cell death, so as to achieve bactericidal effect)" generated through the photothermal effect of the composite hydrogel…”
Authors should unify citing style ( Author at al. [] X Author et al []) and used abbreviations, e.g., PNIPAM and pNIPAM
Response: Thank you for your thoughtful feedback. We apologize for the inconsistency in the citing style and the use of abbreviations in the manuscript. In the revised manuscript, we have carefully reviewed and unified the abbreviations according to the journal's formatting requirements. We have also used abbreviations consistently and appropriately. Thank you for your valuable suggestions. If you have any further suggestions or concerns, please feel free to let us know. Thank you once again for your time and thoughtful evaluation of our work.

Round 2
Reviewer 1 Report
The authors have responded excellently to all of my comments, making the review acceptable for publication in its present form.
Minor editing of English language required.
Reviewer 2 Report
Thanks to the authors for improving the manuscript following the comments and suggestions. I recommend acceptance in this current form. Congrats!